# A scaled Bregman theorem with applications

**Richard Nock**[†,‡,§]     **Aditya Krishna Menon**[†,‡]     **Cheng Soon Ong**[†,‡]
[†]Data61, [‡]the Australian National University and [§]the University of Sydney
{richard.nock, aditya.menon, chengsoon.ong}@data61.csiro.au

## Abstract

Bregman divergences play a central role in the design and analysis of a range of machine learning algorithms through a handful of popular theorems. We present a new theorem which shows that "Bregman distortions" (employing a potentially *non-convex* generator) may be *exactly* re-written as a scaled Bregman divergence computed over transformed data. This property can be viewed from the standpoints of geometry (a scaled isometry with adaptive metrics) or convex optimization (relating generalized perspective transforms). Admissible distortions include geodesic distances on curved manifolds and projections or gauge-normalisation.

Our theorem allows one to leverage to the wealth and convenience of Bregman divergences when analysing algorithms relying on the aforementioned Bregman distortions. We illustrate this with three novel applications of our theorem: a reduction from multi-class density ratio to class-probability estimation, a new adaptive projection free yet norm-enforcing dual norm mirror descent algorithm, and a reduction from clustering on flat manifolds to clustering on curved manifolds. Experiments on each of these domains validate the analyses and suggest that the scaled Bregman theorem might be a worthy addition to the popular handful of Bregman divergence properties that have been pervasive in machine learning.

## 1   Introduction: Bregman divergences as a reduction tool

Bregman divergences play a central role in the design and analysis of a range of machine learning (ML) algorithms. In recent years, Bregman divergences have arisen in procedures for convex optimisation [4], online learning [9, Chapter 11] clustering [3], matrix approximation [13], class-probability estimation [7, 26, 29, 28], density ratio estimation [35], boosting [10], variational inference [18], and computational geometry [5]. Despite these being very different applications, many of these algorithms and their analyses basically rely on three beautiful analytic properties of Bregman divergences, properties that we summarize for differentiable scalar convex functions $\varphi$ with derivative $\varphi'$, conjugate $\varphi^\star$, and divergence $D_\varphi$:

- the triangle equality: $D_\varphi(x\|y) + D_\varphi(y\|z) - D_\varphi(x\|z) = (\varphi'(z) - \varphi'(y))(x - y)$;
- the dual symmetry property: $D_\varphi(x\|y) = D_{\varphi^\star}(\varphi'(y)\|\varphi'(x))$;
- the right-centroid (population minimizer) is the average: $\arg\min_\mu \mathbb{E}[D_\varphi(\mathsf{X}\|\mu)] = \mathbb{E}[\mathsf{X}]$.

Casting a problem as a Bregman minimisation allows one to employ these properties to simplify analysis; for example, by interpreting mirror descent as applying a particular Bregman regulariser, Beck and Teboulle [4] relied on the triangle equality above to simplify its proof of convergence.

Another intriguing possibility is that one may derive *reductions* amongst learning problems by connecting their underlying Bregman minimisations. Menon and Ong [24] recently established how (binary) density ratio estimation (DRE) can be exactly reduced to class-probability estimation (CPE). This was facilitated by interpreting CPE as a Bregman minimisation [7, Section 19], and a new property of Bregman divergences — Menon and Ong [24, Lemma 2] showed that for *any* twice

| Problem A | Problem B that Theorem 1 reduces A to | Reference |
|---|---|---|
| Multiclass density-ratio estimation | Multiclass class-probability estimation | §3, Lemma 2 |
| Online optimisation on $L_q$ ball | Convex unconstrained online learning | §4, Lemma 4 |
| Clustering on curved manifolds | Clustering on flat manifolds | §5, Lemma 5 |

Table 1: Applications of our scaled Bregman Theorem (Theorem 1) — "Reduction" encompasses shortcuts on algorithms *and* on analyses (algorithm/proof A uses algorithm/proof B as subroutine).

differentiable scalar convex $\varphi$, for $g(x) = 1 + x$ and $\check{\varphi}(x) \doteq g(x) \cdot \varphi(x/g(x))$,

$$g(x) \cdot D_\varphi(x/g(x)\|y/g(y)) = D_{\check{\varphi}}(x\|y) \ , \forall x, y. \tag{1}$$

Since the binary class-probability function $\eta(\boldsymbol{x}) = \Pr(\mathsf{Y} = 1|\mathsf{X} = \boldsymbol{x})$ is related to the class-conditional density ratio $r(\boldsymbol{x}) = \Pr(\mathsf{X} = \boldsymbol{x}|\mathsf{Y} = 1)/\Pr(\mathsf{X} = \boldsymbol{x}|\mathsf{Y} = -1)$ via Bayes' rule as $\eta(\boldsymbol{x}) = r(\boldsymbol{x})/g(r(\boldsymbol{x}))$ ([24] assume $\Pr(\mathsf{Y} = 1) = 1/2$), any $\hat{\eta}$ with small $D_\varphi(\eta\|\hat{\eta})$ implicitly produces an $\hat{r}$ with low $D_{\check{\varphi}}(r\|\hat{r})$ i.e. a good estimate of the density ratio. The Bregman property of eq. (1) thus establishes a reduction from DRE to CPE. Two questions arise from this analysis: can we generalise eq. (1) to other $g(\cdot)$, and if so, can we similarly relate *other* problems to each other?

This paper presents a new Bregman identity (Theorem 1), the *scaled Bregman theorem*, a significant generalisation of Menon and Ong [24, Lemma 2]. It shows that general *distortions* $D_{\check{\varphi}}$ – which are not necessarily convex, positive, bounded or symmetric – may be re-expressed as a Bregman divergence $D_\varphi$ computed over transformed data, and thus inherit their good properties despite appearing *prima facie* to be a very different object. This transformation can be as simple as a projection or normalisation by a gauge, or more involved like the exponential map on lifted coordinates for a curved manifold. Our theorem can be summarized in two ways. The first is geometric as it specializes to a scaled isometry involving adaptive metrics. The second calls to a fundamental object of convex analysis, generalized *perspective* transforms [11, 22, 23]. Indeed, our theorem states when

"*the perspective of a Bregman divergence equals the distortion of a perspective*",

for a perspective ($\check{\varphi}$ in eq. 1) which is analytically a generalized perspective transform but does not rely on the same convexity and sign requirements as in Maréchal [22, 23]. We note that the perspective of a Bregman divergence (the left-hand side of eq. 1) is a special case of conformal divergence [27], yet to our knowledge it has never been formally defined. As with the aforementioned key properties of Bregman divergences, Theorem 1 has potentially wide implications for ML. We give three such novel applications to vastly different problems (see Table 1):

- a reduction of multiple density ratio estimation to multiclass-probability estimation (§3), generalising the results of [24] for the binary label case,

- a *projection-free* yet norm-enforcing mirror gradient algorithm (enforced norms are those of mirrored vectors *and* of the offset) with guarantees for adaptive filtering (§4), and

- a seeding approach for clustering on positively or negatively (constant) curved manifolds based on a popular seeding for flat manifolds and with the same approximation guarantees (§5).

Experiments on each of these domains (§6) validate our analysis. The Supplementary Material (SM) details the proofs of all results, provides the experimental results *in extenso* and some additional (nascent) applications to exponential families and computational information geometry.

## 2 Main result: the scaled Bregman theorem

In the remaining, $[k] \doteq \{0, 1, ..., k\}$ and $[k]_* \doteq \{1, 2, ..., k\}$ for $k \in \mathbb{N}$. For any differentiable (but not necessarily convex) $\varphi : \mathcal{X} \to \mathbb{R}$, we define the Bregman *distortion* $D_\varphi$ as

$$D_\varphi(\boldsymbol{x}\|\boldsymbol{y}) \ \doteq \ \varphi(\boldsymbol{x}) - \varphi(\boldsymbol{y}) - (\boldsymbol{x} - \boldsymbol{y})^\top \nabla\varphi(\boldsymbol{y}) \ . \tag{2}$$

If $\varphi$ is convex, $D_\varphi$ is the familiar Bregman *divergence* with generator $\varphi$. Without further ado, we present our main result.

**Theorem 1** *Let, $\varphi : \mathcal{X} \to \mathbb{R}$ be convex differentiable, and $g : \mathcal{X} \to \mathbb{R}_*$ be differentiable. Then,*

$$g(\boldsymbol{x}) \cdot D_\varphi \left( (1/g(\boldsymbol{x})) \cdot \boldsymbol{x} \ \| \ (1/g(\boldsymbol{y})) \cdot \boldsymbol{y} \right) \ = \ D_{\check{\varphi}} \left( \boldsymbol{x} \ \| \ \boldsymbol{y} \right) \ , \forall \boldsymbol{x}, \boldsymbol{y} \in \mathcal{X} \ , \tag{3}$$

$$\textit{where } \check{\varphi}(\boldsymbol{x}) \ \doteq \ g(\boldsymbol{x}) \cdot \varphi \left( (1/g(\boldsymbol{x})) \cdot \boldsymbol{x} \right) \ , \tag{4}$$

| $\mathcal{X}$ | $D_\varphi(\boldsymbol{x}\|\boldsymbol{y})$ | $D_{\check\varphi}(\boldsymbol{x}\|\boldsymbol{y})$ | $g(\boldsymbol{x})$ |
|---|---|---|---|
| $\mathbb{R}^d$ | $\frac{1}{2}\cdot\|\boldsymbol{x}-\boldsymbol{y}\|_2^2$ | $\|\boldsymbol{x}\|_2\cdot(1-\cos\angle\boldsymbol{x},\boldsymbol{y})$ | $\|\boldsymbol{x}\|_2$ |
| $\mathbb{R}^d$ | $\frac{1}{2}\cdot(\|\boldsymbol{x}\|_q^2-\|\boldsymbol{y}\|_q^2)-\sum_i\frac{(x_i-y_i)\cdot\text{sign}(y_i)\cdot|y_i|^{q-1}}{\|\boldsymbol{y}\|_q^{q-2}}$ | $W\cdot\|\boldsymbol{x}\|_q-W\cdot\sum_i\frac{x_i\cdot\text{sign}(y_i)\cdot|y_i|^{q-1}}{\|\boldsymbol{y}\|_q^{q-1}}$ | $\|\boldsymbol{x}\|_q/W$ |
| $\mathbb{R}^d\times\mathbb{R}$ | $\frac{1}{2}\|\boldsymbol{x}^S-\boldsymbol{y}^S\|_2^2$ | $\frac{\|\boldsymbol{x}\|_2}{\sin\|\boldsymbol{x}\|_2}\cdot(1-\cos D_G(\boldsymbol{x},\boldsymbol{y}))$ | $\|\boldsymbol{x}\|_2/\sin\|\boldsymbol{x}\|_2$ |
| $\mathbb{R}^d\times\mathbb{C}$ | $\frac{1}{2}\|\boldsymbol{x}^H-\boldsymbol{y}^H\|_2^2$ | $-\frac{\|\boldsymbol{x}\|_2}{\sinh\|\boldsymbol{x}\|_2}\cdot(\cosh D_G(\boldsymbol{x},\boldsymbol{y})-1)$ | $-\|\boldsymbol{x}\|_2/\sinh\|\boldsymbol{x}\|_2$ |
| $\mathbb{R}^d_+$ | $\sum_i x_i\log\frac{x_i}{y_i}-\mathbf{1}^\top(\boldsymbol{x}-\boldsymbol{y})$ | $\sum_i x_i\log\frac{x_i}{y_i}-d\cdot\mathbb{E}[X]\cdot\log\frac{\mathbb{E}[X]}{\mathbb{E}[Y]}$ | $\mathbf{1}^\top\boldsymbol{x}$ |
| $\mathbb{R}^d_+$ | $\sum_i\frac{x_i}{y_i}-\sum_i\log\frac{x_i}{y_i}-d$ | $\sum_i\frac{x_i(\prod_j y_j)^{1/d}}{y_i}-d(\prod_j x_j)^{1/d}$ | $\prod_i x_i^{1/d}$ |
| $\mathbf{S}(d)$ | $\text{tr}(\text{x}\log\text{x}-\text{x}\log\text{y})-\text{tr}(\text{x})+\text{tr}(\text{y})$ | $\text{tr}(\text{x}\log\text{x}-\text{x}\log\text{y})-\text{tr}(\text{x})\cdot\log\frac{\text{tr}(\text{X})}{\text{tr}(\text{Y})}$ | $\text{tr}(\text{x})$ |
| $\mathbf{S}(d)$ | $\text{tr}(\text{XY}^{-1})-\log\det(\text{XY}^{-1})-d$ | $\det(\text{Y}^{1/d})\text{tr}(\text{XY}^{-1})-d\cdot\det(\text{X}^{1/d})$ | $\det(\text{X}^{1/d})$ |

Table 2: Examples of $(D_\varphi, D_{\check\varphi}, g)$ for which eq. (3) holds. Function $\boldsymbol{x}^S\doteq f(\boldsymbol{x}):\mathbb{R}^d\to\mathbb{R}^{d+1}$ and $\boldsymbol{x}^H\doteq f(\boldsymbol{x}):\mathbb{R}^d\to\mathbb{R}^d\times\mathbb{C}$ are the Sphere and Hyperbolic lifting maps defined in SM, eqs. 51, 62. $W>0$ is a constant. $D_G$ denotes the *G*eodesic distance on the sphere (for $\boldsymbol{x}^S$) or the hyperboloid (for $\boldsymbol{x}^H$). $\mathbf{S}(d)$ is the set of symmetric real matrices. Related proofs are in SM, Section III.

*if and only if (i) g is affine on $\mathcal{X}$, or (ii) for every $\boldsymbol{z}\in\mathcal{X}_g\doteq\{(1/g(\boldsymbol{x}))\cdot\boldsymbol{x}:\boldsymbol{x}\in\mathcal{X}\}$,*

$$\varphi(\boldsymbol{z})=\boldsymbol{z}^\top\nabla\varphi(\boldsymbol{z}) \ . \tag{5}$$

Table 2 presents some examples of (sometimes involved) triplets $(D_\varphi, D_{\check\varphi}, g)$ for which eq. (3) holds; related proofs are in Appendix III. Depending on $\varphi$ and $g$, there are at least two ways to summarize Theorem 1. One is geometric: Theorem 1 sometimes states a *scaled isometry* between $\mathcal{X}$ and $\mathcal{X}_g$. The other one comes from convex optimisation: Theorem 1 defines *generalized perspective transforms* on Bregman divergences and roughly states the identity between the perspective transform of a Bregman divergence and the Bregman distortion of the perspective transform. Appendix VIII gives more details for both properties. We refer to Theorem 1 as the *scaled Bregman theorem*.

**Remark.** If $\mathcal{X}_g$ is a vector space, $\varphi$ satisfies eq. (5) if and only if it is positive homogeneous of degree 1 on $\mathcal{X}_g$ (i.e. $\varphi(\alpha\boldsymbol{z})=\alpha\cdot\varphi(\boldsymbol{z})$ for any $\alpha>0$) from Euler's homogenous function theorem. When $\mathcal{X}_g$ is not a vector space, this only holds for $\alpha$ such that $\alpha\boldsymbol{z}\in\mathcal{X}_g$ as well. We thus call the gradient condition of eq. (5) "restricted positive homogeneity" for simplicity. ∎

**Remark.** Appendix IV gives a "deep composition" extension of Theorem 1. ∎

For the special case where $\mathcal{X}=\mathbb{R}$, and $g(x)=1+x$, Theorem 1 is exactly [24, Lemma 2] (c.f. eq. 1). We wish to highlight a few points with regard to our more general result. First, the "distortion" generator $\check\varphi$ may be[1] *non-convex*, as the following illustrates.

**Example.** Suppose $\varphi(\boldsymbol{x})=(1/2)\|\boldsymbol{x}\|_2^2$, the generator for squared Euclidean distance. Then, for $g(\boldsymbol{x})=1+\mathbf{1}^\top\boldsymbol{x}$, we have $\check\varphi(\boldsymbol{x})=(1/2)\cdot\|\boldsymbol{x}\|_2^2/(1+\mathbf{1}^\top\boldsymbol{x})$, which is non-convex on $\mathcal{X}=\mathbb{R}^d$. ∎

When $\check\varphi$ is non-convex, the right hand side in eq. (3) is an object that ostensibly bears only a superficial similarity to a Bregman divergence; it is somewhat remarkable that Theorem 1 shows this general "distortion" between a pair $(\boldsymbol{x},\boldsymbol{y})$ to be entirely equivalent to a (scaling of a) Bregman divergence between some transformation of the points. Second, when $g$ is linear, eq. (3) holds for *any* convex $\varphi$ (This was the case considered in [24]). When $g$ is non-linear, however, $\varphi$ must be chosen carefully so that $(\varphi, g)$ satisfies the restricted homogeneity conditon[2] of eq. (5). In general, given a convex $\varphi$, one can "reverse engineer" a suitable $g$, as illustrated by the following example.

**Example.** Suppose[3] $\varphi(\boldsymbol{x})=(1+\|\boldsymbol{x}\|_2^2)/2$. Then, eq. (5) requires that $\|\boldsymbol{x}\|_2^2=1$ for every $\boldsymbol{x}\in\mathcal{X}_g$, i.e. $\mathcal{X}_g$ is (a subset of) the unit sphere. This is afforded by the choice $g(\boldsymbol{x})=\|\boldsymbol{x}\|_2$. ∎

Third, Theorem 1 is not merely a mathematical curiosity: we now show that it facilitates novel results in three very different domains, namely estimating multiclass density ratios, constrained online optimisation, and clustering data on a manifold with non-zero curvature. We discuss nascent applications to exponential families and computational geometry in Appendices V and VI.

# 3 Multiclass density-ratio estimation via class-probability estimation

Given samples from a number of densities, density ratio estimation concerns estimating the ratio between each density and some reference density. This has applications in the covariate shift problem wherein the train and test distributions over instances differ [33]. Our first application of Theorem 1 is to show how density ratio estimation can be reduced to class-probability estimation [7, 29].

To proceed, we fix notation. For some integer $C \geq 1$, consider a distribution $\mathbb{P}(\mathsf{X}, \mathsf{Y})$ over an (instance, label) space $\mathcal{X} \times [C]$. Let $(\{P_c\}_{c=1}^{C}, \boldsymbol{\pi})$ be densities giving $\mathbb{P}(\mathsf{X}|\mathsf{Y} = c)$ and $\mathbb{P}(\mathsf{Y} = c)$ respectively, and $M$ giving $\mathbb{P}(\mathsf{X})$ accordingly. Fix $c^* \in [C]$ a reference class, and suppose for simplicity that $c^* = C$. Let $\tilde{\boldsymbol{\pi}} \in \triangle^{C-1}$ such that $\tilde{\pi}_c \doteq \pi_c/(1 - \pi_C)$. *Density ratio estimation* [35] concerns inferring the vector $\boldsymbol{r}(\boldsymbol{x}) \in \mathbb{R}^{C-1}$ of density ratios relative to $C$, with $r_c(\boldsymbol{x}) \doteq \mathbb{P}(\mathsf{X} = \boldsymbol{x}|\mathsf{Y} = c)/\mathbb{P}(\mathsf{X} = \boldsymbol{x}|\mathsf{Y} = C)$ , while *class-probability estimation* [7] concerns inferring the vector $\boldsymbol{\eta}(\boldsymbol{x}) \in \mathbb{R}^{C-1}$ of class-probabilities, with $\eta_c(\boldsymbol{x}) \doteq \mathbb{P}(\mathsf{Y} = c|\mathsf{X} = \boldsymbol{x})/\tilde{\pi}_c$ . In both cases, we estimate the respective quantities given an iid sample $\mathcal{S} \sim \mathbb{P}(\mathsf{X}, \mathsf{Y})^m$ ($m$ is the training sample size).

The genesis of the reduction from density ratio to class-probability estimation is the fact that $\boldsymbol{r}(\boldsymbol{x}) = (\pi_C/(1 - \pi_C)) \cdot \boldsymbol{\eta}(\boldsymbol{x})/\eta_C(\boldsymbol{x})$. In practice one will only have an estimate $\hat{\boldsymbol{\eta}}$, typically derived by minimising a suitable loss on the given $\mathcal{S}$ [37], with a canonical example being multiclass logistic regression. Given $\hat{\boldsymbol{\eta}}$, it is natural to estimate the density ratio via:

$$\hat{\boldsymbol{r}}(\boldsymbol{x}) = \hat{\boldsymbol{\eta}}(\boldsymbol{x})/\hat{\eta}_C(\boldsymbol{x}) \ . \tag{6}$$

While this estimate is intuitive, to establish a formal reduction we must relate the quality of $\hat{\boldsymbol{r}}$ to that of $\hat{\boldsymbol{\eta}}$. Since the minimisation of a suitable loss for class-probability estimation is equivalent to a Bregman minimisation [7, Section 19], [37, Proposition 7], this is however immediate by Theorem 1:

**Lemma 2** *Given a class-probability estimator $\hat{\eta} \colon \mathcal{X} \to [0, 1]^{C-1}$, let the density ratio estimator $\hat{r}$ be as per Equation 6. Then for any convex differentiable $\varphi \colon [0, 1]^{C-1} \to \mathbb{R}$,*

$$\mathbb{E}_{\mathsf{X} \sim M}[D_\varphi(\boldsymbol{\eta}(\mathsf{X}) \| \hat{\boldsymbol{\eta}}(\mathsf{X}))] = (1 - \pi_C) \cdot \mathbb{E}_{\mathsf{X} \sim P_C} \left[ D_{\varphi^\dagger}(\boldsymbol{r}(\mathsf{X}) \| \hat{\boldsymbol{r}}(\mathsf{X})) \right] \tag{7}$$

*where $\varphi^\dagger$ is as per Equation 4 with $g(\boldsymbol{x}) \doteq \pi_C/(1 - \pi_C) + \tilde{\boldsymbol{\pi}}^\top \boldsymbol{x}$ .*

Lemma 2 generalises [24, Proposition 3], which focussed on the binary case with $\pi = 1/2$ (See Appendix VII for a review of that result). Unpacking the Lemma, the LHS in Equation 7 represents the object minimised by some suitable loss for class-probability estimation. Since $g$ is affine, we can use *any* convex, differentiable $\varphi$, and so can use *any* suitable class-probability loss to estimate $\hat{\boldsymbol{\eta}}$. Lemma 2 thus implies that producing $\hat{\boldsymbol{\eta}}$ by minimising any class-probability loss *equivalently* produces an $\hat{\boldsymbol{r}}$ as per Equation 6 that minimises a Bregman divergence to the true $\boldsymbol{r}$. Thus, Theorem 1 provides a reduction from density ratio to multiclass probability estimation.

We now detail two applications where $g(\cdot)$ is no longer affine, and $\varphi$ must be chosen more carefully.

# 4 Dual norm mirror descent: projection-free online learning on $L_p$ balls

A substantial amount of work in the intersection of ML and convex optimisation has focused on constrained optimisation within a ball [32, 14]. This optimisation is typically via projection operators that can be expensive to compute [17, 19]. We now show that *gauge functions* can be used as an inexpensive alternative, and that Theorem 1 easily yields guarantees for this procedure in online learning. We consider the adaptive filtering problem, closely related to the online least squares problem with linear predictors [9, Chapter 11]. Here, over a sequence of $T$ rounds, we observe some $\boldsymbol{x}_t \in \mathcal{X}$. We must then predict a target value $\hat{y}_t = \boldsymbol{w}_{t-1}^\top \boldsymbol{x}_t$ using our current weight vector $\boldsymbol{w}_{t-1}$. The true target $y_t = \boldsymbol{u}^\top \boldsymbol{x}_t + \epsilon_t$ is then revealed, where $\epsilon_t$ is some unknown noise, and we may update our weight to $\boldsymbol{w}_t$. Our goal is to minimise the *regret* of the sequence $\{\boldsymbol{w}_t\}_{t=0}^{T}$,

$$R(\boldsymbol{w}_{1:T}|\boldsymbol{u}) \doteq \sum_{t=1}^{T} \left( \boldsymbol{u}^\top \boldsymbol{x}_t - \boldsymbol{w}_{t-1}^\top \boldsymbol{x}_t \right)^2 - \sum_{t=1}^{T} \left( \boldsymbol{u}^\top \boldsymbol{x}_t - y_t \right)^2 \ . \tag{8}$$

Let $q \in (1, 2]$ and $p$ be such that $1/p + 1/q = 1$. For $\varphi \doteq (1/2) \cdot \|\boldsymbol{x}\|_q^2$ and loss $\ell_t(\boldsymbol{w}) = (1/2) \cdot (y_t - \boldsymbol{w}^\top \boldsymbol{x}_t)^2$, the $p$-LMS algorithm [20] employs the stochastic mirror gradient updates:

$$\boldsymbol{w}_t \doteq \operatorname*{argmin}_{\boldsymbol{w}} \eta_t \cdot \ell_t(\boldsymbol{w}) + D_\varphi(\boldsymbol{w} \| \boldsymbol{w}_{t-1}) = (\nabla\varphi)^{-1} \left( \nabla\varphi(\boldsymbol{w}_{t-1}) - \eta_t \cdot \nabla\ell_t \right), \tag{9}$$

where $\eta_t$ is a learning rate to be specified by the user. [20, Theorem 2] shows that for appropriate $\eta_t$, one has $R(\boldsymbol{w}_{1:T}|\boldsymbol{u}) \leq (p-1) \cdot \max_{\boldsymbol{x} \in \mathcal{X}} \|\boldsymbol{x}\|_p^2 \cdot \|\boldsymbol{u}\|_q^2$.

The $p$-LMS updates do not provide any explicit control on $\|\boldsymbol{w}_t\|$, i.e. there is no regularisation. Experiments (Section §6) suggest that leaving $\|\boldsymbol{w}_t\|$ uncontrolled may not be a good idea as the increase of the norm sometimes prevents (significant) updates (eq. (9)). Also, the wide success of regularisation in ML calls for regularised variants that *retain* the regret guarantees and computational efficiency of $p$-LMS. (Adding a projection step to eq. (9) would not achieve both.) We now do just this. For fixed $W > 0$, let $\varphi \doteq (1/2) \cdot (W^2 + \|\boldsymbol{x}\|_q^2)$, a translation of that used in $p$-LMS. Invoking Theorem 1 with the admissible $g_q(\boldsymbol{x}) = \|\boldsymbol{x}\|_q/W$ yields $\check{\varphi} = \check{\varphi}_q = W\|\boldsymbol{x}\|_q$ (see Table 2). Using the fact that $L_p$ and $L_q$ norms are dual of each other, we replace eq. (9) by:

$$\boldsymbol{w}_t \doteq \nabla\check{\varphi}_p \left( \nabla\check{\varphi}_q(\boldsymbol{w}_{t-1}) - \eta_t \cdot \nabla\ell_t \right) \ . \tag{10}$$

See Lemma A of the Appendix for the simple forms of $\nabla\check{\varphi}_{\{p,q\}}$. We call update (10) the *dual norm p-LMS (DN-p-LMS) algorithm*, noting that the dual refers to the polar transform of the norm, and $g$ stems from a gauge normalization for $\mathcal{B}_q(W)$, the closed $L_q$ ball with radius $W > 0$. Namely, we have $\gamma_{\text{GAU}}(\boldsymbol{x}) = W/\|\boldsymbol{x}\|_q = g(\boldsymbol{x})^{-1}$ for the gauge $\gamma_{\text{GAU}}(\boldsymbol{x}) \doteq \sup\{z \geq 0 : z \cdot \boldsymbol{x} \in \mathcal{B}_q(W)\}$, so that $\check{\varphi}_q$ implicitly performs gauge normalisation of the data. This update is no more computationally expensive than eq. (9) — we simply need to compute the $p$- and $q$-norms of appropriate terms — but, crucially, automatically constrains the norms of $\boldsymbol{w}_t$ and its image by $\nabla\check{\varphi}_q$.

**Lemma 3** *For the update in eq. (10), $\|\boldsymbol{w}_t\|_q = \|\nabla\check{\varphi}_q(\boldsymbol{w}_t)\|_p = W, \forall t > 0$.*

Lemma 3 is remarkable, since *nowhere in eq. (10) do we project onto the $L_q$ ball*. Nonetheless, for the DN-$p$-LMS updates to be principled, we need a similar regret guarantee to the original $p$-LMS. Fortunately, this may be done using Theorem 1 to exploit the original proof of [20]. For any $\boldsymbol{u} \in \mathbb{R}^d$, define the *q-normalised regret* of $\{\boldsymbol{w}_t\}_{t=0}^T$ by

$$R_q(\boldsymbol{w}_{1:T}|\boldsymbol{u}) \doteq \sum_{t=1}^T \left( (1/g_q(\boldsymbol{u})) \cdot \boldsymbol{u}^\top \boldsymbol{x}_t - \boldsymbol{w}_{t-1}^\top \boldsymbol{x}_t \right)^2 - \sum_{t=1}^T \left( (1/g_q(\boldsymbol{u})) \cdot \boldsymbol{u}^\top \boldsymbol{x}_t - y_t \right)^2 \tag{11}$$

We have the following bound on $R_q$ for the DN-$p$-LMS updates (We cannot expect a bound on the unnormalised $R(\cdot)$ of eq. (8), since by Lemma 3 we can only compete against norm $W$ vectors).

**Lemma 4** *Pick any $\boldsymbol{u} \in \mathbb{R}^d$, $p,q$ satisfying $1/p + 1/q = 1$ and $p > 2$, and $W > 0$. Suppose $\|\boldsymbol{x}_t\|_p \leq X_p$ and $|y_t| \leq Y, \forall t \leq T$. Let $\{\boldsymbol{w}_t\}$ be as per eq. (10), using learning rate*

$$\eta_t \doteq \gamma_t \cdot \frac{W}{4(p-1)\max\{W, X_p\}X_pW + |y_t - \boldsymbol{w}_{t-1}^\top \boldsymbol{x}_t|X_p} \ , \tag{12}$$

*for **any** desired $\gamma_t \in [1/2, 1]$. Then,*

$$R_q(\boldsymbol{w}_{1:T}|\boldsymbol{u}) \leq 4(p-1)X_p^2W^2 + (16p - 8)\max\{W, X_p\}X_p^2W + 8YX_p^2 \ . \tag{13}$$

Several remarks can be made. First, the bound depends on the maximal signal value $Y$, but this is the maximal signal in the observed sequence, so it may not be very large in practice; if it is comparable to $W$, then our bound is looser than [20] by just a constant factor. Second, the learning rate is adaptive in the sense that its choice depends on the last mistake made. There is a nice way to represent the "offset" vector $\eta_t \cdot \nabla\ell_t$ in eq. (10), since we have, for $Q'' \doteq 4(p-1)\max\{W, X_p\}X_pW$,

$$\eta_t \cdot \nabla\ell_t = W \cdot \frac{|y_t - \boldsymbol{w}_{t-1}^\top \boldsymbol{x}_t|X_p}{Q'' + |y_t - \boldsymbol{w}_{t-1}^\top \boldsymbol{x}_t|X_p} \cdot \text{sign}(y_t - \boldsymbol{w}_{t-1}^\top \boldsymbol{x}_t) \cdot \left( \frac{1}{X_p} \cdot \boldsymbol{x} \right) \ , \tag{14}$$

so the $L_p$ norm of the offset is actually equal to $W \cdot Q$, where $Q \in [0,1]$ is all the smaller as the vector $\boldsymbol{w}_.$ gets better. Hence, the update in eq. (10) controls in fact *all* norms (that of $\boldsymbol{w}_.$, its image by $\nabla\check{\varphi}_q$ and the offset). Third, because of the normalisation of $\boldsymbol{u}$, the bound actually does not depend on $\boldsymbol{u}$, but on the radius $W$ chosen for the $L_q$ ball.

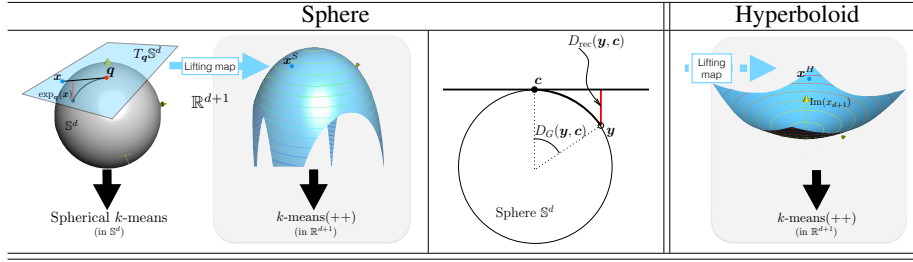

Figure 1: (L) Lifting map into $\mathbb{R}^d \times \mathbb{R}$ for clustering on the sphere with k-means++. (M) $D_{\text{rec}}$ in Eq. (15) in vertical thick red line. (R) Lifting map into $\mathbb{R}^d \times \mathbb{C}$ for the hyperboloid.

## 5    Clustering on a curved manifold via clustering on a flat manifold

Our final application can be related to two problems that have received a steadily growing interest over the past decade in unsupervised ML: clustering on a non-linear manifold [12], and subspace custering [36]. We consider two fundamental manifolds investigated by [16] to compute centers of mass from relativistic theory: the sphere $\mathbb{S}^d$ and the hyperboloid $\mathbb{H}^d$, the former being of positive curvature, and the latter of negative curvature. Applications involving these specific manifolds are numerous in text processing, computer vision, geometric modelling, computer graphics, to name a few [8, 12, 15, 21, 30, 34]. We emphasize the fact that the clustering problem has significant practical impact for $d$ as small as 2 in computer vision [34].

The problem is non-trivial for two separate reasons. First, the ambient space, *i.e.* the space of registration of the input data, is often implicitly Euclidean and therefore *not* the manifold [12]: if the mapping to the manifold is not carefully done, then geodesic distances measured on the manifold may be inconsistent with respect to the ambient space. Second, the fact that the manifold has non-zero curvature essentially prevents the direct use of Euclidean optimization algorithms [38] — put simply, the average of two points that belong to a manifold does not necessarily belong to the manifold, so we have to be careful on how to compute centroids for hard clustering [16, 27, 30, 31].

What we show now is that Riemannian manifolds with constant sectional curvature may be clustered with the $k$-means++ seeding for flat manifolds [2], *without even touching a line of the algorithm*. To formalise the problem, we need three key components of Riemannian geometry: tangent planes, exponential map and geodesics [1]. We assume that the ambient space is a tangent plane to the manifold $\mathcal{M}$, which conveniently makes it look Euclidean (see Figure 1). The point of tangency is called $\boldsymbol{q}$, and the tangent plane $T_{\boldsymbol{q}}\mathcal{M}$. The exponential map, $\exp_{\boldsymbol{q}} : T_{\boldsymbol{q}}\mathcal{M} \to \mathcal{M}$, performs a distance preserving mapping: the geodesic length between $\boldsymbol{q}$ and $\exp_{\boldsymbol{q}}(\boldsymbol{x})$ in $\mathcal{M}$ is the same as the Euclidean length between $\boldsymbol{q}$ and $\boldsymbol{x}$ in $T_{\boldsymbol{q}}\mathcal{M}$. Our clustering objective is to find $\mathcal{C} \doteq \{\boldsymbol{c}_1, \boldsymbol{c}_2, ...\boldsymbol{c}_k\} \subset \mathcal{M}$ such that $D_{\text{rec}}(\mathcal{S} : \mathcal{C}) = \inf_{\mathcal{C}' \subset \mathcal{M}, |\mathcal{C}'|=k} D_{\text{rec}}(\mathcal{S}, \mathcal{C}')$, with

$$D_{\text{rec}}(\mathcal{S}, \mathcal{C}) \quad \doteq \sum_{i \in [m]_*} \min_{j \in [k]_*} D_{\text{rec}}(\exp_{\boldsymbol{q}}(\boldsymbol{x}_i), \boldsymbol{c}_j) \ , \tag{15}$$

where $D_{\text{rec}}$ is a *rec*onstruction loss, a function of the geodesic distance between $\exp_{\boldsymbol{q}}(\boldsymbol{x}_i)$ and $\boldsymbol{c}_j$. We use two loss functions defined from [16] and used in ML for more than a decade [12]:

$$\mathbb{R}_+ \ni D_{\text{rec}}(\boldsymbol{y}, \boldsymbol{c}) \quad \doteq \quad \begin{cases} 1 - \cos D_G(\boldsymbol{y}, \boldsymbol{c}) & \text{for} & \mathcal{M} = \mathbb{S}^d \\ \cosh D_G(\boldsymbol{y}, \boldsymbol{c}) - 1 & \text{for} & \mathcal{M} = \mathbb{H}^d \end{cases} . \tag{16}$$

Here, $D_G(\boldsymbol{y}, \boldsymbol{c})$ is the corresponding geodesic distance of $\mathcal{M}$ between $\boldsymbol{y}$ and $\boldsymbol{c}$. Figure 1 shows that $D_{\text{rec}}(\boldsymbol{y}, \boldsymbol{c})$ is the orthogonal distance between $T_{\boldsymbol{c}}\mathcal{M}$ and $\boldsymbol{y}$ when $\mathcal{M} = \mathbb{S}^d$. The solution to the clustering problem in eq. (15) is therefore the one that minimizes the error between tangent planes defined at the centroids, and points on the manifold.

It turns out that both distances in 16 can be engineered as Bregman divergences via Theorem 1, as seen in Table 2. Furthermore, they imply the same $\varphi$, which is just the generator of Mahalanobis distortion, but a different $g$. The construction involves a third party, a *lifting map* (lift(.)) that increases the dimension by one. The *Sphere* lifting map $\mathbb{R}^d \ni \boldsymbol{x} \mapsto \boldsymbol{x}^S \in \mathbb{R}^{d+1}$ is indicated in Table 3 (left). The new coordinate depends on the norm of $\boldsymbol{x}$. The *Hyperbolic* lifting map, $\mathbb{R}^d \ni \boldsymbol{x} \mapsto \boldsymbol{x}^H \in \mathbb{R}^d \times \mathbb{C}$, involves a pure imaginary additional coordinate, is indicated in in Table 3 (right, with a slight abuse of notation) and Figure 1. Both $\boldsymbol{x}^S$ and $\boldsymbol{x}^H$ live on a $d$-dimensional manifold, depicted in Figure 1.

| (Sphere) S$k$-means++$(\mathcal{S}, k)$ | (Hyperboloid) H$k$-means++$(\mathcal{S}, k)$ |
|---|---|
| **Input:** dataset $\mathcal{S} \subset T_{\boldsymbol{q}}\mathbb{S}^d, k \in \mathbb{N}_*$; | **Input:** dataset $\mathcal{S} \subset T_{\boldsymbol{q}}\mathbb{H}^d, k \in \mathbb{N}_*$; |
| Step 1: $\mathcal{S}^+ \leftarrow \{g_S^{-1}(\boldsymbol{x}^S) \cdot \boldsymbol{x}^S : \boldsymbol{x}^S \in \text{lift}(\mathcal{S})\}$; | Step 1: $\mathcal{S}^+ \leftarrow \{g_H^{-1}(\boldsymbol{x}^H) \cdot \boldsymbol{x}^H : \boldsymbol{x}^H \in \text{lift}(\mathcal{S})\}$; |
| Step 2: $\mathcal{C}^+ \leftarrow k\text{-means++\_seeding}(\mathcal{S}^+, k)$; | Step 2: $\mathcal{C}^+ \leftarrow k\text{-means++\_seeding}(\mathcal{S}^+, k)$; |
| Step 3: $\mathcal{C} \leftarrow \exp_{\boldsymbol{q}}^{-1}(\mathcal{C}^+)$; | Step 3: $\mathcal{C} \leftarrow \exp_{\boldsymbol{q}}^{-1}(\mathcal{C}^+)$; |
| **Output:** Cluster centers $\mathcal{C} \in T_{\boldsymbol{q}}\mathbb{S}^d$; | **Output:** Cluster centers $\mathcal{C} \in T_{\boldsymbol{q}}\mathbb{H}^d$; |
| $\boldsymbol{x}^S \doteq [x_1 \quad x_2 \quad \cdots \quad x_d \quad \|\boldsymbol{x}\|_2 \cot \|\boldsymbol{x}\|_2]$ | $\boldsymbol{x}^H \doteq [x_1 \quad x_2 \quad \cdots \quad x_d \quad i\|\boldsymbol{x}\|_2 \coth \|\boldsymbol{x}\|_2]$ |
| $g_S(\boldsymbol{x}^S) \doteq \|\boldsymbol{x}\|_2 / \sin\|\boldsymbol{x}\|_2$ | $g_H(\boldsymbol{x}^H) \doteq -\|\boldsymbol{x}\|_2 / \sinh\|\boldsymbol{x}\|_2$ |

Table 3: How to use $k$-means++ to cluster points on the sphere (left) or the hyperboloid (right).

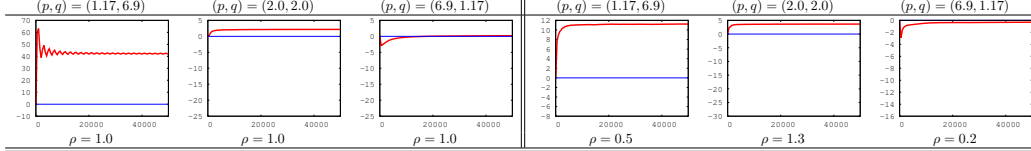

Table 4: Summary of the experiments displaying ($y$) the error of $p$-LMS minus error of DN-$p$-LMS (when $> 0$, DN-$p$-LMS beats $p$-LMS) as a function of $t$, in the setting of [20], for various values of $(p, q)$ (columns). Left panel: (D)ense target; Right panel: (S)parse target.

When they are scaled by the corresponding $g_\cdot(.)$, they happen to be mapped to $\mathbb{S}^d$ or $\mathbb{H}^d$, respectively, by what happens to be the manifold's exponential map for the original $\boldsymbol{x}$ (see Appendix III).

Theorem 1 is interesting in this case because $\varphi$ corresponds to a Mahalanobis distortion: this shows that $k$-means++ seeding [2, 25] can be used directly on the scaled coordinates $(g_{\{S,H\}}^{-1}(\boldsymbol{x}^{\{S,H\}}) \cdot \boldsymbol{x}^{\{S,H\}})$ to pick centroids that yield an approximation of the global optimum for the clustering problem on the manifold which is just *as good as* the original Euclidean approximation bound [2].

**Lemma 5** *The expected potential of S$k$-means++ seeding over the random choices of $\mathcal{C}^+$ satisfies:*

$$\mathbb{E}[D_{\text{rec}}(\mathcal{S} : \mathcal{C})] \quad \leq \quad 8(2 + \log k) \cdot \inf_{\mathcal{C}' \in \mathbb{S}^d} D_{\text{rec}}(\mathcal{S} : \mathcal{C}') \ . \tag{17}$$

*The same approximation bounds holds for H$k$-means++ seeding on the hyperboloid ($\mathcal{C}', \mathcal{C}^+ \in \mathbb{H}^d$).*

Lemma 5 is notable since it was only recently shown that such a bound is possible for the sphere [15], and to our knowledge, no such approximation quality is known for clustering on the hyperboloid [30, 31]. Notice that Lloyd iterations on non-linear manifolds would require repetitive renormalizations to keep centers on the manifold [12], an additional disadvantage compared to clustering on flat manifolds that $\{G, K\}$-means++ seedings do not bear.

## 6 Experimental validation

We present some experiments validating our theoretical analysis for the applications above.

**Multiple density ratio estimation**. See Appendix IX for experiments in this domain.

**Dual norm $p$-LMS (DN-$p$-LMS)**. We ran $p$-LMS and the DN-$p$-LMS of §4 on the experimental setting of [20]. We refer to that paper for an exhaustive description of the experimental setting, which we briefly summarize: it is a noisy signal processing setting, involving a dense or a sparse target. We compute, over the signal received, the error of our predictor on the signal. We keep all parameters as they are in [20], except for one: we make sure that data are scaled to fit in a $L_p$ ball of prescribed radius, to test the assumption related in [20] that fixing the learning rate $\eta_t$ is not straightforward in $p$-LMS. Knowing the true value of $X_p$, we then scale it by a misestimation factor $\rho$, typically in $[0.1, 1.7]$. We use the same misestimation in DN-$p$-LMS. Thus, both algorithms suffer the same source of uncertainty. Also, we periodically change the signal (each 1000 iterations), to assess the performances of the algorithms in tracking changes in the signal.

Experiments, given *in extenso* in Appendix X, are sumarized in Table 4. The following trends emerge: in the mid to long run, DN-$p$-LMS is never beaten by $p$-LMS by more than a fraction of percent. On the other hand, DN-$p$-LM can beat $p$-LMS by very significant differences (exceeding 40%), in particular when $p < 2$, *i.e.* when we are outside the regime of the proof of [20]. This indicates that

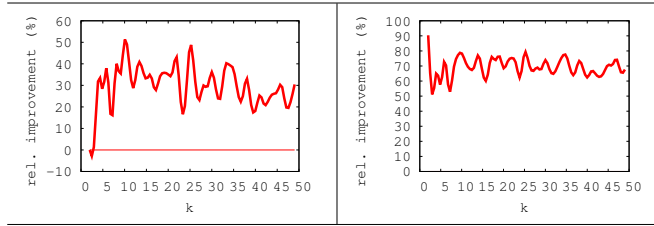

Table 5: (L) Relative improvement (decrease) in $k$-means potential of SKM∘S$k$-means++ compared to SKM alone. (R) Relative improvement of S$k$-means++ over Forgy initialization on the sphere.

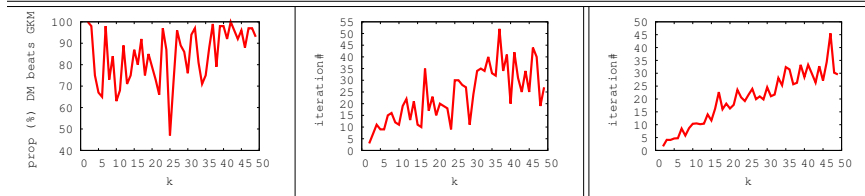

Table 6: (L) % of the number of runs of SKM whose output (*when it has converged*) is better than S$k$-means++. (C) Maximal # of iterations for SKM after which it beats S$k$-means++ (ignoring runs of SKM that do not beat S$k$-means++). (R) Average # of iterations for SKM to converge.

significantly stronger and more general results than the one of Lemma 4 may be expected. Also, it seems that the problem of $p$-LMS lies in an "exploding" norm problem: in various cases, we observe that $\|\boldsymbol{w}_t\|$ (in any norm) blows up with $t$, and this correlates with a very significant degradation of its performances. Clearly, DN-$p$-LMS does not have this problem since all relevant norms are under tight control. Finally, even when the norm does not explode, DN-$p$-LMS can still beat $p$-LMS, by less important differences though. Of course, the output of $p$-LMS can repeatedly be normalised, but the normalisation would escape the theory of [20] and it is not clear which normalisation would be best.

**Clustering on the sphere**. For $k \in [50]_*$, we simulate on $T_{\mathbf{0}}\mathbb{S}^2$ a mixture of spherical Gaussian and uniform densities in random rectangles with $2k$ components. We run three algorithms: (i) SKM [12] on the data embedded on $\mathbb{S}^2$ with random (Forgy) initialization, (ii), S$k$-means++ and (iii) SKM with S$k$-means++ initialisation. Results are averaged over the algorithms' runs.

Table 5 (left) displays that using S$k$-means++ as initialization for SKM brings a very significant gain over SKM alone, since we almost divide the $k$-means potential by a factor 2 on some runs. The right plot of Table 5 shows that S-$k$-means++ consistently reduces the $k$-means potential by at least a factor 2 over Forgy. The left plot in Table 6 displays that even when it has converged, SKM does *not* necessarily beat S$k$-means++. Finally, the center+right plots in Table 6 display that even when it does beat S$k$-means++ when it has converged, the iteration number after which SKM beats S$k$-means++ increases with $k$, and in the worst case may *exceed* the average number of iterations needed for SKM to converge (we stopped SKM if relative improvement is not above $1‰$).

## 7 Conclusion

We presented a new scaled Bregman identity, and used it to derive novel results in several fields of machine learning: multiple density ratio estimation, adaptive filtering, and clustering on curved manifolds. We believe that, like other known key properties of Bregman divergences, there is potential for other applications of the result; Appendices V, VI present preliminary thoughts in this direction.

## 8 Acknowledgments

The authors wish to thank Bob Williamson and the reviewers for insightful comments.

## Footnotes

[1]Evidently, $\check\varphi$ is convex iff $g$ is non-negative, by eq. (3) and the fact that a function is convex iff its Bregman "distortion" is nonnegative [6, Section 3.1.3].

[2]We stress that this condition only needs to hold on $\mathcal{X}_g\subseteq\mathcal{X}$; it would not be really interesting in general for $\varphi$ to be homogeneous *everywhere* in its domain, since we would basically have $\check\varphi=\varphi$.

[3]The constant $1/2$ added in $\varphi$ does *not* change $D_\varphi$, since a Bregman divergence is invariant to affine terms; removing this however would make the divergences $D_\varphi$ and $D_{\check\varphi}$ differ by a constant.

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
