[Supplementary Material · nips16-nmo-camera-ready-sm.pdf]

# A scaled Bregman theorem with applications
# — Supplementary Material —

### Abstract

This is the Supplementary Material to Paper "A scaled Bregman theorem with applications" by R. Nock, A-K. Menon and C.-S. Ong. Theorems and Lemmata are numbered with letters (A, B, ...) to make a clear difference with the main file numbering.

## Table of contents

# I  Proofs of results in main body

We present proofs of all results in the main body.

**Proof** [Proof of Theorem 1] Let $J : \mathcal{X} \to \mathcal{X}_g$ denote the Jacobian of $h\colon \boldsymbol{x} \mapsto (1/g(\boldsymbol{x})) \cdot \boldsymbol{x}$. By an elementary calculation,

$$g(\boldsymbol{x}) \cdot J = I_d - (1/g(\boldsymbol{x})) \cdot \boldsymbol{x} \nabla g(\boldsymbol{x})^\top,$$

which by the chain rule brings the following expression for the gradient of $\check{\varphi}(\boldsymbol{y}) = g(\boldsymbol{y}) \cdot (\varphi \circ h)(\boldsymbol{y})$:

$$
\begin{aligned}
\nabla \check{\varphi}(\boldsymbol{y}) &= \nabla g(\boldsymbol{y}) \cdot (\varphi \circ h)(\boldsymbol{y}) + g(\boldsymbol{y}) \cdot \nabla(\varphi \circ h)(\boldsymbol{y}) \\
&= \nabla g(\boldsymbol{y}) \cdot (\varphi \circ h)(\boldsymbol{y}) + g(\boldsymbol{y}) \cdot J^\top \nabla \varphi(h(\boldsymbol{y})) \\
&= \nabla g(\boldsymbol{y}) \cdot (\varphi \circ h)(\boldsymbol{y}) + \nabla \varphi(h(\boldsymbol{y})) - (1/g(\boldsymbol{y})) \cdot \nabla g(\boldsymbol{y}) \boldsymbol{y}^\top \nabla \varphi(h(\boldsymbol{y})) \\
&= \nabla \varphi\left(\frac{1}{g(\boldsymbol{y})} \cdot \boldsymbol{y}\right) + \left(\varphi\left(\frac{1}{g(\boldsymbol{y})} \cdot \boldsymbol{y}\right) - \frac{1}{g(\boldsymbol{y})} \cdot \boldsymbol{y}^\top \nabla \varphi\left(\frac{1}{g(\boldsymbol{y})} \cdot \boldsymbol{y}\right)\right) \cdot \nabla g(\boldsymbol{y}) \ . \quad (1)
\end{aligned}
$$

For simplicity, let $\boldsymbol{u} = \boldsymbol{x}/g(\boldsymbol{x})$ and $\boldsymbol{v} = \boldsymbol{y}/g(\boldsymbol{y})$, so that $\check{\varphi}(\boldsymbol{x}) = g(\boldsymbol{x}) \cdot \varphi(\boldsymbol{u})$ and $\check{\varphi}(\boldsymbol{y}) = g(\boldsymbol{y}) \cdot \varphi(\boldsymbol{v})$. The above then reads

$$\nabla \check{\varphi}(\boldsymbol{y}) = \nabla \varphi(\boldsymbol{v}) + \left(\varphi(\boldsymbol{v}) - \boldsymbol{v}^\top \nabla \varphi(\boldsymbol{v})\right) \cdot \nabla g(\boldsymbol{y}). \quad (2)$$

Now, the LHS of Equation (3) (main file) is

$$
\begin{aligned}
g(\boldsymbol{x}) \cdot D_\varphi\left(\frac{1}{g(\boldsymbol{x})} \cdot \boldsymbol{x} \;\middle\|\; \frac{1}{g(\boldsymbol{y})} \cdot \boldsymbol{y}\right) &= g(\boldsymbol{x}) \cdot D_\varphi(\boldsymbol{u}\|\boldsymbol{v}) \\
&= g(\boldsymbol{x}) \cdot \varphi(\boldsymbol{u}) - g(\boldsymbol{x}) \cdot \varphi(\boldsymbol{v}) - g(\boldsymbol{x}) \cdot \nabla \varphi(\boldsymbol{v})^\top(\boldsymbol{u} - \boldsymbol{v}) \\
&= \check{\varphi}(\boldsymbol{x}) - g(\boldsymbol{x}) \cdot \varphi(\boldsymbol{v}) - \nabla \varphi(\boldsymbol{v})^\top(\boldsymbol{x} - g(\boldsymbol{x}) \cdot \boldsymbol{v}) \\
&= \check{\varphi}(\boldsymbol{x}) - g(\boldsymbol{x}) \cdot (\varphi(\boldsymbol{v}) - \nabla \varphi(\boldsymbol{v})^\top \boldsymbol{v}) - \nabla \varphi(\boldsymbol{v})^\top \boldsymbol{x},
\end{aligned}
$$

while the RHS is

$$
\begin{aligned}
D_{\check{\varphi}}(\boldsymbol{x} \;\|\; \boldsymbol{y}) &= \check{\varphi}(\boldsymbol{x}) - \check{\varphi}(\boldsymbol{y}) - \nabla \check{\varphi}(\boldsymbol{y})^\top(\boldsymbol{x} - \boldsymbol{y}) \\
&= \check{\varphi}(\boldsymbol{x}) - g(\boldsymbol{y}) \cdot \varphi(\boldsymbol{v}) - \nabla \varphi(\boldsymbol{v})^\top(\boldsymbol{x} - \boldsymbol{y}) - \left(\varphi(\boldsymbol{v}) - \boldsymbol{v}^\top \nabla \varphi(\boldsymbol{v})\right) \cdot \nabla g(\boldsymbol{y})^\top(\boldsymbol{x} - \boldsymbol{y}).
\end{aligned}
$$

Cancelling the common $\check{\varphi}(\boldsymbol{x})$ and $\nabla \varphi(\boldsymbol{v})^\top \boldsymbol{y}$ terms, the difference $\Delta = \mathrm{RHS} - \mathrm{LHS}$ is

$$
\begin{aligned}
\Delta &= g(\boldsymbol{x}) \cdot (\varphi(\boldsymbol{v}) - \nabla \varphi(\boldsymbol{v})^\top \boldsymbol{v}) - g(\boldsymbol{y}) \cdot \varphi(\boldsymbol{v}) + \nabla \varphi(\boldsymbol{v})^\top \boldsymbol{y} \\
&\quad - \left(\varphi(\boldsymbol{v}) - \boldsymbol{v}^\top \nabla \varphi(\boldsymbol{v})\right) \cdot \nabla g(\boldsymbol{y})^\top(\boldsymbol{x} - \boldsymbol{y}) \\
&= g(\boldsymbol{x}) \cdot (\varphi(\boldsymbol{v}) - \nabla \varphi(\boldsymbol{v})^\top \boldsymbol{v}) - g(\boldsymbol{y}) \cdot \varphi(\boldsymbol{v}) + g(\boldsymbol{y}) \cdot \nabla \varphi(\boldsymbol{v})^\top \boldsymbol{v} \\
&\quad - \left(\varphi(\boldsymbol{v}) - \boldsymbol{v}^\top \nabla \varphi(\boldsymbol{v})\right) \cdot \nabla g(\boldsymbol{y})^\top(\boldsymbol{x} - \boldsymbol{y}) \\
&= g(\boldsymbol{x}) \cdot (\varphi(\boldsymbol{v}) - \nabla \varphi(\boldsymbol{v})^\top \boldsymbol{v}) - g(\boldsymbol{y}) \cdot \left(\varphi(\boldsymbol{v}) - \nabla \varphi(\boldsymbol{v})^\top \boldsymbol{v}\right) \\
&\quad - \left(\varphi(\boldsymbol{v}) - \boldsymbol{v}^\top \nabla \varphi(\boldsymbol{v})\right) \cdot \nabla g(\boldsymbol{y})^\top(\boldsymbol{x} - \boldsymbol{y}) \\
&= (\varphi(\boldsymbol{v}) - \nabla \varphi(\boldsymbol{v})^\top \boldsymbol{v}) \cdot (g(\boldsymbol{x}) - g(\boldsymbol{y}) - \nabla g(\boldsymbol{y})^\top(\boldsymbol{x} - \boldsymbol{y})) \\
&= (\varphi(\boldsymbol{v}) - \nabla \varphi(\boldsymbol{v})^\top \boldsymbol{v}) \cdot D_g(\boldsymbol{x}\|\boldsymbol{y}).
\end{aligned}
$$

Thus, the identity holds, if and only if either $\varphi(\boldsymbol{v}) = \nabla\varphi(\boldsymbol{v})^\top \boldsymbol{v}$ for every $\boldsymbol{v} \in \mathcal{X}_g$, or $D_g(\boldsymbol{x}\|\boldsymbol{y}) = 0$. The latter is true if and only if $g$ is affine from Equation 2. The result follows. ∎

It is easy to check that Theorem 1 in fact holds for separable (matrix) trace divergences [Kulis et al., 2009] of the form

$$D_\varphi(\text{X}\|\text{Y}) \;\dot{=}\; \varphi(\text{X}) - \varphi(\text{Y}) - \operatorname{tr}\left(\nabla\varphi(\text{Y})^\top(\text{X} - \text{Y})\right) \;, \tag{3}$$

with $\varphi, g : \mathbf{S}(d) \to \mathbb{R}$ (for $\mathbf{S}(d)$ the set of symmetric real matrices), with $\varphi$ convex. In this case, the restricted positive homogeneity property becomes

$$\varphi(\text{U}) \;=\; \operatorname{tr}\left(\nabla\varphi(\text{U})^\top \text{U}\right) \;, \forall\text{U} \in \mathcal{X}_g \;. \tag{4}$$

**Proof** [Proof of Lemma 2] Note that by construction, $g(\boldsymbol{r}(\boldsymbol{x})) = \mathbb{P}(\text{X} = \boldsymbol{x})/((1 - \pi_C) \cdot \mathbb{P}(\text{X} = \boldsymbol{x}|\text{Y} = C))$, and so

$$
\begin{aligned}
\left(\frac{1}{g(\boldsymbol{r}(\boldsymbol{x}))} \cdot \boldsymbol{r}(\boldsymbol{x})\right)_c &= \frac{(1 - \pi_C) \cdot \mathbb{P}(\text{X} = \boldsymbol{x}|\text{Y} = C)}{\mathbb{P}(\text{X} = \boldsymbol{x})} \cdot \frac{\mathbb{P}(\text{X} = \boldsymbol{x}|\text{Y} = c)}{\mathbb{P}(\text{X} = \boldsymbol{x}|\text{Y} = C)} \\
&= \frac{(1 - \pi_C)}{\pi_c} \cdot \frac{\pi_c \mathbb{P}(\text{X} = \boldsymbol{x}|\text{Y} = c)}{\mathbb{P}(\text{X} = \boldsymbol{x})} \\
&= \eta(\boldsymbol{x}) \;.
\end{aligned}
\tag{5}
$$

Furthermore,

$$
\begin{aligned}
\mathbb{P}(\text{X} = \boldsymbol{x}) &= \sum_{c=1}^C \pi_c \mathbb{P}(\text{X} = \boldsymbol{x}|\text{Y} = c) \\
&= (1 - \pi_C) \cdot \left(\frac{\pi_C}{1 - \pi_C} + \sum_{c<C} \frac{\pi_c}{1 - \pi_C} \cdot \frac{\mathbb{P}(\text{X} = \boldsymbol{x}|\text{Y} = c)}{\mathbb{P}(\text{X} = \boldsymbol{x}|\text{Y} = C)}\right) \cdot \mathbb{P}(\text{X} = \boldsymbol{x}|\text{Y} = C) \\
&= (1 - \pi_C) \cdot g(\boldsymbol{r}(\boldsymbol{x})) \cdot \mathbb{P}(\text{X} = \boldsymbol{x}|\text{Y} = C) \;.
\end{aligned}
\tag{6}
$$

Now let

$$\hat{\boldsymbol{r}}(\boldsymbol{x}) = \frac{1}{\hat{\eta}_C(\boldsymbol{x})} \cdot \hat{\boldsymbol{\eta}}(\boldsymbol{x}).$$

It then comes

$$
\begin{aligned}
\mathbb{E}_M&[D_\varphi(\boldsymbol{\eta}(\text{X})\|\hat{\boldsymbol{\eta}}(\text{X}))] \\
&= (1 - \pi_C) \cdot \mathbb{E}_{P_C}\left[g(\boldsymbol{r}(\boldsymbol{x})) \cdot D_\varphi(\boldsymbol{\eta}(\text{X})\|\hat{\boldsymbol{\eta}}(\text{X}))\right] \\
&= (1 - \pi_C) \cdot \mathbb{E}_{P_C}\left[g(\boldsymbol{r}(\boldsymbol{x})) \cdot D_\varphi\left(\frac{1}{g(\boldsymbol{r}(\boldsymbol{x}))} \cdot \boldsymbol{r}(\boldsymbol{x}) \,\middle\|\, \hat{\boldsymbol{\eta}}(\text{X})\right)\right] \\
&= (1 - \pi_C) \cdot \mathbb{E}_{P_C}\left[g(\boldsymbol{r}(\boldsymbol{x})) \cdot D_\varphi\left(\frac{1}{g(\boldsymbol{r}(\boldsymbol{x}))} \cdot \boldsymbol{r}(\boldsymbol{x}) \,\middle\|\, \frac{1}{g(\hat{\boldsymbol{r}}(\text{X}))} \cdot \hat{\boldsymbol{r}}(\text{X})\right)\right] \\
&= (1 - \pi_C) \cdot \mathbb{E}_{P_C}\left[D_{\check{\varphi}}(\boldsymbol{r}(\text{X})\|\hat{\boldsymbol{r}}(\text{X}))\right] \;,
\end{aligned}
$$

as claimed. ∎

**Proof** [Proof of Lemma 3] For any $\boldsymbol{x}$, $\|\nabla\check{\varphi}_p(\boldsymbol{x})\|_q = W$ by Corollary B. Since $\boldsymbol{w}_t = \nabla\check{\varphi}_p(\boldsymbol{\theta}_{t-1})$ for suitable $\boldsymbol{\theta}_{t-1}$, the result follows. The result for $\|\nabla\check{\varphi}_q(\boldsymbol{w}_t)\|_p$ follows similarly by Corollary B.

Note that while $\|\boldsymbol{w}_t\|_q = \|\nabla\varphi(\boldsymbol{w}_t)\|_p$ for the standard $p$-LMS update [Kivinen et al., 2006, Appendix I], these norms may vary with each iteration i.e. $\boldsymbol{w}_t$ may not lie in the $L_q$ ball. ∎

**Proof** [Proof of Lemma 4] Similarly to the proof of Lemma E, a key to the proof of Lemma 4 relies on branching on Kivinen et al. [2006] through the use of Theorem 1. We first note that $D_{\check{\varphi}_q}(\boldsymbol{u}\|\boldsymbol{w}_0) = W \cdot \|\boldsymbol{u}\|_q$ since $\boldsymbol{w}_0 = \boldsymbol{0}$, and $D_{\check{\varphi}_q}(\boldsymbol{u}\|\boldsymbol{w}_{T+1}) \geq 0$, and so

$$
\begin{aligned}
W \cdot \|\boldsymbol{u}\|_q \quad \geq \quad & D_{\check{\varphi}_q}(\boldsymbol{u}\|\boldsymbol{w}_0) - D_{\check{\varphi}_q}(\boldsymbol{u}\|\boldsymbol{w}_{T+1}) \\
= \quad & \sum_{t=1}^{T} \left\{ D_{\check{\varphi}_q}(\boldsymbol{u}\|\boldsymbol{w}_{t-1}) - D_{\check{\varphi}_q}(\boldsymbol{u}\|\boldsymbol{w}_t) \right\} \text{ by telescoping property} \\
= \quad & g_q(\boldsymbol{u}) \cdot \sum_{t=1}^{T} \left\{ D_{\varphi_q}\left( \frac{\boldsymbol{u}}{g_q(\boldsymbol{u})} \left\| \frac{\boldsymbol{w}_{t-1}}{g_q(\boldsymbol{w}_{t-1})} \right.\right) - D_{\varphi_q}\left( \frac{\boldsymbol{u}}{g_q(\boldsymbol{u})} \left\| \frac{\boldsymbol{w}_t}{g_q(\boldsymbol{w}_t)} \right.\right) \right\} \text{ by Theorem 1} \\
= \quad & g_q(\boldsymbol{u}) \cdot \sum_{t=1}^{T} \left\{ D_{\varphi_q}\left( \frac{\boldsymbol{u}}{g_q(\boldsymbol{u})} \| \boldsymbol{w}_{t-1} \right) - D_{\varphi_q}\left( \frac{\boldsymbol{u}}{g_q(\boldsymbol{u})} \| \boldsymbol{w}_t \right) \right\} \text{ by Lemma J} \ . \qquad (7)
\end{aligned}
$$

Recall from Lemma I that

$$
D_{\varphi_q}\left( \frac{\boldsymbol{u}}{g_q(\boldsymbol{u})} \| \boldsymbol{w}_{t-1} \right) - D_{\varphi_q}\left( \frac{\boldsymbol{u}}{g_q(\boldsymbol{u})} \| \boldsymbol{w}_t \right) \geq \frac{1}{4(p-1)\left(2 + \frac{4M}{W} + \frac{2(Y+X_pW)}{(p-1)W^2}\right)X_p^2} \cdot (s_t^2 - r_t^2)
$$

where

$$
\begin{aligned}
s_t &\doteq ((1/g_q(\boldsymbol{u})) \cdot \boldsymbol{u} - \boldsymbol{w}_{t-1})^\top \boldsymbol{x}_t \\
r_t &\doteq (1/g_q(\boldsymbol{u})) \cdot \boldsymbol{u}^\top \boldsymbol{x}_t - y_t.
\end{aligned}
$$

Note that $R_q(\boldsymbol{w}_{1:T}|\boldsymbol{u}) = \sum_{t=1}^{T}(s_t^2 - r_t^2)$ by definition. Summing the above for $t = 1, 2, ..., T$ and telescoping sums yields

$$
\begin{aligned}
R_q(\boldsymbol{w}_{1:T}|\boldsymbol{u}) \quad \leq \quad & 4(p-1)\left(2 + \frac{4M}{W} + \frac{2(Y+X_pW)}{(p-1)W^2}\right)X_p^2 W^2 \\
= \quad & 4(p-1)X_p^2 W^2 + 16(p-1)M X_p^2 W + 8(Y + X_pW)X_p^2 \\
\leq \quad & 4(p-1)X_p^2 W^2 + (16p-8)M X_p^2 W + 8Y X_p^2 \ . \qquad (8)
\end{aligned}
$$

See Figure 1 for some geometric intuition about the updates. ∎

Figure 1: Illustration of the case $W = 1$ for the $\mathcal{B}_q(W)$-update: all classifiers and image via $\nabla\check{\varphi}$ belong to a ball of radius 1 (here, $q = 3$, $p = 3/2$).

**Proof** [Proof of Lemma 5] We start by the sphere. Let $\varphi(\boldsymbol{x}) \doteq (1/2) \cdot \|\boldsymbol{x}\|_2^2$. Since a Bregman divergence is invariant to linear transformation, it comes from Table A1 that

$$D_\varphi\left(\frac{\boldsymbol{x}^S}{g_S(\boldsymbol{x}^S)} \,\Big\|\, \frac{\boldsymbol{c}^S}{g_S(\boldsymbol{c}^S)}\right) = \frac{1}{g_S(\boldsymbol{c}^S)} \cdot D_{\check{\varphi}}(\boldsymbol{x}\|\boldsymbol{c}) = 1 - \cos D_G(\boldsymbol{x}, \boldsymbol{c}),$$

where we recall that $D_G$ denotes the geodesic distance on the sphere (see Figure 1 and Appendix III). Equivalently,

$$\left\| \frac{1}{g_S(\boldsymbol{x}^S)} \cdot \boldsymbol{x}^S - \frac{1}{g_S(\boldsymbol{c}^S)} \cdot \boldsymbol{c}^S \right\|_2^2 = 1 - \cos D_G(\boldsymbol{x}, \boldsymbol{c}) \ . \tag{9}$$

This equality allows us to use $k$-means++ using the LHS of (9) to compute the distribution that picks a center. The key to using the approximation property of $k$-means++ relies on the existence of a coordinate system on the sphere for which the cluster centroid is just the average of the cluster points (polar coordinates), an average that eventually has to be rescaled if the coordinate system is not that one [Dhillon and Modha, 2001, Endo and Miyamoto, 2015]. The existence of this coordinate system makes that the proof of Arthur and Vassilvitskii [2007] (and in particular the key Lemmata 3.2 and 3.3) can be carried out without modification to yield the same approximation ratio as that of Arthur and Vassilvitskii [2007] *if* the distortion at hand is the squared Euclidean distance, which turns out to be $D_{\text{rec}}(.:.)$ from eq. (9).

The case of the hyperboloid follows the exact same path, but starts from the fact that Table A1 now brings

$$D_\varphi\left(\frac{\boldsymbol{x}^H}{g_H(\boldsymbol{x}^H)} \,\Big\|\, \frac{\boldsymbol{c}^H}{g_H(\boldsymbol{c}^H)}\right) = \cosh D_G(\boldsymbol{y}, \boldsymbol{c}) - 1 = \left\| \frac{1}{g_H(\boldsymbol{x}^H)} \cdot \boldsymbol{x}^H - \frac{1}{g_H(\boldsymbol{c}^H)} \cdot \boldsymbol{c}^H \right\|_2^2 \ .$$

To finish, in the same way as for the Sphere, we just need the existence of a coordinate system for which the centroid is an average of the cluster points, which can be obtained from hyperbolic barycentric coordinates [Ungar, 2014, Section 18]. ∎

# II   Additional helper lemmata

We begin with some helper lemmata that will be used in some of the proofs. In what follows, let

$$\varphi_q(\boldsymbol{w}) = (1/2)(W^2 + \|\boldsymbol{w}\|_q^2)$$
$$\check{\varphi}_q(\boldsymbol{w}) = W \cdot \|\boldsymbol{w}\|_q$$

for some $W > 0$ and $p, q \in (1, \infty)$ such that $1/p + 1/q = 1$.

## II.1   Properties of $\varphi_q$ and $\check{\varphi}_q$

We use the following properties of $\varphi_q, \check{\varphi}_q$.

**Lemma A**  *For any $\boldsymbol{w}$,*

$$\nabla \varphi_q(\boldsymbol{w}) = \|\boldsymbol{w}\|_q^{2-q} \cdot \text{sign}(\boldsymbol{w}) \otimes |\boldsymbol{w}|^{q-1}$$
$$\nabla \check{\varphi}_q(\boldsymbol{w}) = W \cdot \|\boldsymbol{w}\|_q^{1-q} \cdot \text{sign}(\boldsymbol{w}) \otimes |\boldsymbol{w}|^{q-1},$$

*where $\otimes$ denotes Hadamard product.*

**Proof**  The first identity was shown in [Kivinen et al., 2006, Example 1]. The second identity follows from a simple calculation. ∎

This implies the following useful relations between the gradients of $\varphi_q$ and $\check{\varphi}_q$.

**Corollary B**  *For any $\boldsymbol{w}$,*

$$\nabla \varphi_q(\boldsymbol{w}) = (\|\boldsymbol{w}\|_q / W) \cdot \nabla \check{\varphi}_q(\boldsymbol{w}),$$
$$\|\nabla \check{\varphi}_q(\boldsymbol{w})\|_p = W,$$
$$\|\nabla \varphi_q(\boldsymbol{w})\|_p = \|\boldsymbol{w}\|_q.$$

**Proof**  [Proof of Corollary B] The proof follows by direct application of Lemma A and the definition of $p, q$. Note the third identity was shown in Kivinen et al. [2006, Appendix I]. ∎

As a consequence, we conclude that the gradients of $\varphi_q$ and $\check{\varphi}_q$ coincide when considering vectors on the $W$-sphere.

**Lemma C**  *For any $\|\boldsymbol{w}\|_q = W$,*
$$\nabla \varphi_q(\boldsymbol{w}) = \nabla \check{\varphi}_q(\boldsymbol{w}).$$

**Proof**  This follows from the relation between $\nabla \varphi_q$ and $\nabla \check{\varphi}_q$ from Lemma A. ∎

Finally, we have the following result about the composition of gradients.

**Lemma D**  *For any $\boldsymbol{w}$,*

$$\nabla \varphi_q \circ \nabla \check{\varphi}_p(\boldsymbol{w}) = \nabla \check{\varphi}_q \circ \nabla \check{\varphi}_p(\boldsymbol{w}) = \frac{W}{\|\boldsymbol{w}\|_p} \cdot \boldsymbol{w}.$$

**Proof** For the first identity, applying Lemma A twice,

$$
\begin{aligned}
\nabla \varphi_q \circ \nabla \check{\varphi}_p(\boldsymbol{w}) \;=\;& \frac{1}{\|\nabla \check{\varphi}_p(\boldsymbol{w})\|_q^{q-2}} \cdot \operatorname{sign}(\nabla \check{\varphi}_p(\boldsymbol{w})) \otimes |\nabla \check{\varphi}_p(\boldsymbol{w})|^{q-1} \\
=\;& \frac{1}{W_q^{q-2}} \cdot \operatorname{sign}(\boldsymbol{w}) \otimes \frac{W^{q-1}}{\|\boldsymbol{w}\|_p^{(p-1)(q-1)}} \cdot |\boldsymbol{w}|^{(p-1)(q-1)} \\
=\;& \frac{W}{\|\boldsymbol{w}\|_p} \cdot \boldsymbol{w} \ .
\end{aligned}
\tag{10}
$$

For the second identity, use Corollary B to conclude that

$$
\begin{aligned}
\nabla \check{\varphi}_q \circ \nabla \check{\varphi}_p(\boldsymbol{w}) =\;& \frac{W}{\|\nabla \check{\varphi}_p(\boldsymbol{w})\|_q} \cdot \nabla \varphi_q(\nabla \check{\varphi}_p(\boldsymbol{w})) \\
=\;& W \cdot \frac{\boldsymbol{w}}{\|\boldsymbol{w}\|_p},
\end{aligned}
$$

as claimed. ∎

## II.2   Bound on successive iterate divergence

The following Lemma extends [Kivinen et al., 2006, Appendix I] to $\check{\varphi}_q$.

**Lemma E**  *For any $\boldsymbol{w}$ and $\boldsymbol{\delta}$,*

$$
\begin{aligned}
& D_{\check{\varphi}_q} \left( \boldsymbol{w} \| \nabla \check{\varphi}_p \left( \nabla \check{\varphi}_q(\boldsymbol{w}) + \boldsymbol{\delta} \right) \right) \\
& \leq\; \frac{(p-1)\|\boldsymbol{w}\|_q W}{2} \cdot \left\| \frac{1}{\|\nabla \check{\varphi}_q(\boldsymbol{w}) + \boldsymbol{\delta}\|_p} \cdot (\nabla \check{\varphi}_q(\boldsymbol{w}) + \boldsymbol{\delta}) - \frac{1}{W} \cdot \nabla \check{\varphi}_q(\boldsymbol{w}) \right\|_p^2 \ .
\end{aligned}
\tag{11}
$$

**Proof** [Proof of Lemma E] In this proof, $\circ$ denotes composition and $\otimes$ is Hadamard product. The key step in the proof is the use of Theorem 1 to "branch" on the proof of [Kivinen et al., 2006, Appendix I] on the first following identity (letting $\varphi_q(\boldsymbol{w}) \doteq (1/2) \cdot (W^2 + \|\boldsymbol{w}\|_q^2)$). We also make use of the dual

symmetry of Bregman divergences and we obtain third identity of:

$$D_{\check{\varphi}_q}\left(\boldsymbol{w}\|\nabla\check{\varphi}_p\left(\nabla\check{\varphi}_q(\boldsymbol{w})+\boldsymbol{\delta}\right)\right)$$

$$= \frac{\|\boldsymbol{w}\|_q}{W}\cdot D_{\varphi_q}\left(\frac{W}{\|\boldsymbol{w}\|_q}\cdot\boldsymbol{w}\left\|\frac{W}{\|\nabla\check{\varphi}_p(\nabla\check{\varphi}_q(\boldsymbol{w})+\boldsymbol{\delta})\|_q}\cdot\nabla\check{\varphi}_p\left(\nabla\check{\varphi}_q(\boldsymbol{w})+\boldsymbol{\delta}\right)\right)\right)$$

$$= \frac{\|\boldsymbol{w}\|_q}{W}\cdot D_{\varphi_q}\left(\frac{W}{\|\boldsymbol{w}\|_q}\cdot\boldsymbol{w}\left\|\nabla\check{\varphi}_p\left(\nabla\check{\varphi}_q(\boldsymbol{w})+\boldsymbol{\delta}\right)\right)\right) \tag{12}$$

$$= \frac{\|\boldsymbol{w}\|_q}{W}\cdot D_{\varphi_p}\left(\nabla\varphi_q\circ\nabla\check{\varphi}_p\left(\nabla\check{\varphi}_q(\boldsymbol{w})+\boldsymbol{\delta}\right)\left\|\nabla\varphi_q\left(\frac{W}{\|\boldsymbol{w}\|_q}\cdot\boldsymbol{w}\right)\right)\right) \quad\text{by dual symmetry}$$

$$= \frac{\|\boldsymbol{w}\|_q}{W}\cdot D_{\varphi_p}\left(\frac{W}{\|\nabla\check{\varphi}_q(\boldsymbol{w})+\boldsymbol{\delta}\|_p}\cdot(\nabla\check{\varphi}_q(\boldsymbol{w})+\boldsymbol{\delta})\left\|\frac{W}{\|\boldsymbol{w}\|_q}\cdot\nabla\varphi_q\left(\boldsymbol{w}\right)\right)\right) \tag{13}$$

$$= \frac{\|\boldsymbol{w}\|_q}{W}\cdot D_{\varphi_p}\left(\frac{W}{\|\nabla\check{\varphi}_q(\boldsymbol{w})+\boldsymbol{\delta}\|_p}\cdot(\nabla\check{\varphi}_q(\boldsymbol{w})+\boldsymbol{\delta})\left\|\nabla\check{\varphi}_q\left(\boldsymbol{w}\right)\right)\right) \tag{14}$$

$$= \|\boldsymbol{w}\|_q W\cdot D_{\varphi_p}\left(\frac{1}{\|\nabla\check{\varphi}_q(\boldsymbol{w})+\boldsymbol{\delta}\|_p}\cdot(\nabla\check{\varphi}_q(\boldsymbol{w})+\boldsymbol{\delta})\left\|\frac{1}{W}\cdot\nabla\check{\varphi}_q\left(\boldsymbol{w}\right)\right)\right)\quad. \tag{15}$$

Equations (12) – (14) hold because of Corollary B. We now use Appendix I[1] in Kivinen et al. [2006] on Equation (15) and obtain

$$D_{\check{\varphi}_q}\left(\boldsymbol{w}\|\nabla\check{\varphi}_p\left(\nabla\check{\varphi}_q(\boldsymbol{w})+\boldsymbol{\delta}\right)\right)$$

$$\leq \frac{(p-1)\|\boldsymbol{w}\|_q W}{2}\cdot\left\|\frac{1}{\|\nabla\check{\varphi}_q(\boldsymbol{w})+\boldsymbol{\delta}\|_p}\cdot(\nabla\check{\varphi}_q(\boldsymbol{w})+\boldsymbol{\delta})-\frac{1}{W}\cdot\nabla\check{\varphi}_q\left(\boldsymbol{w}\right)\right\|_p^2\quad,$$

as claimed. ■

## II.3  Bound on successive iterate divergence to target

In what follows, we write the DN-$p$-LMS updates as $\boldsymbol{w}_t=\nabla\check{\varphi}_p(\boldsymbol{\theta}_t)$, where

$$\boldsymbol{\theta}_t\doteq\nabla\check{\varphi}_q(\boldsymbol{w}_{t-1})-\Delta_t$$

for $\Delta_t=\eta_t\cdot(\boldsymbol{w}_{t-1}^\top\boldsymbol{x}_t-y_t)\cdot\boldsymbol{x}_t$. Further, for notational ease, we write

$$\bar{\boldsymbol{u}}\doteq\frac{\boldsymbol{u}}{g_q(\boldsymbol{u})}$$

and

$$\bar{\boldsymbol{\theta}}_t\doteq\frac{\boldsymbol{\theta}_t}{\|\boldsymbol{\theta}_t\|_p}.$$

We have the following preliminary bound on the distance from iterates of DN-$p$-LMS to the (normalised) target.

**Lemma F** *Fix any learning rate sequence* $\{\eta_t\}_{t=1}^T$. *Pick any* $\boldsymbol{u}$, *and consider iterates* $\{\boldsymbol{w}_t\}_{t=0}^T$ *as per the update equation:*

$$\boldsymbol{w}_t \;\doteq\; \nabla\check{\varphi}_p\left(\nabla\check{\varphi}_q(\boldsymbol{w}_{t-1}) - \eta_t\cdot\nabla\ell_t\right)\;. \tag{16}$$

*Denote* $s_t \doteq (\bar{\boldsymbol{u}} - \boldsymbol{w}_{t-1})^\top\boldsymbol{x}_t, r_t \doteq \bar{\boldsymbol{u}}^\top\boldsymbol{x}_t - y_t$, *and* $\alpha_t \doteq \frac{W}{\|\boldsymbol{\theta}_t\|_p}$. *Suppose* $\|\boldsymbol{x}_t\|_p \le X_p$. *Then,*

$$D_{\varphi_q}\left(\bar{\boldsymbol{u}}\,\|\,\boldsymbol{w}_{t-1}\right) - D_{\varphi_q}\left(\bar{\boldsymbol{u}}\,\|\,\boldsymbol{w}_t\right) \;\ge\; Q + R + S + T\;,$$

*with*

$$
\begin{aligned}
Q &\doteq \frac{\alpha_t}{2}\eta_t(s_t^2 - r_t^2),\\
R &\doteq (1-\alpha_t)\cdot\underbrace{\left(W^2 - \bar{\boldsymbol{u}}^\top\nabla\check{\varphi}_q(\boldsymbol{w}_{t-1})\right)}_{\in[0,2W^2]}\;,\\
S &\doteq \frac{p-1}{2}\cdot\underbrace{\left(2\alpha_t^2\eta_t^2(s_t-r_t)^2X_p^2 - \|(s_t-r_t)\eta_t\alpha_t\cdot\boldsymbol{x}_t - (1-\alpha_t)\cdot\nabla\check{\varphi}_q(\boldsymbol{w}_{t-1})\|_p^2\right)}_{\ge -2(1-\alpha_t)^2 W^2}\;,\\
T &\doteq \frac{\alpha_t}{2}\eta_t(s_t-r_t)^2\left(1 - 2(p-1)\eta_t\alpha_t X_p^2\right)\;.
\end{aligned}
$$

**Proof** [Proof of Lemma F] The Bregman triangle equality (also called the three points property) [Boissonnat et al., 2010, Property 5], [Cesa-Bianchi and Lugosi, 2006, Lemma 11.1] brings:

$$
\begin{aligned}
&D_{\varphi_q}\left(\bar{\boldsymbol{u}}\,\|\,\boldsymbol{w}_{t-1}\right) - D_{\varphi_q}\left(\bar{\boldsymbol{u}}\,\|\,\boldsymbol{w}_t\right)\\
&= (\bar{\boldsymbol{u}} - \boldsymbol{w}_{t-1})^\top\left(\nabla\varphi_q\left(\boldsymbol{w}_t\right) - \nabla\varphi_q\left(\boldsymbol{w}_{t-1}\right)\right) - D_{\varphi_q}\left(\boldsymbol{w}_{t-1}\,\|\,\boldsymbol{w}_t\right)\\
&= (\bar{\boldsymbol{u}} - \boldsymbol{w}_{t-1})^\top\left(\nabla\check{\varphi}_q\left(\boldsymbol{w}_t\right) - \nabla\check{\varphi}_q\left(\boldsymbol{w}_{t-1}\right)\right) - D_{\check{\varphi}_q}\left(\boldsymbol{w}_{t-1}\,\|\,\boldsymbol{w}_t\right)\ \text{ by Lemma 3 (main file) and Lemma C}\;.
\end{aligned}
$$

We now have

$$\nabla\check{\varphi}_q\left(\boldsymbol{w}_t\right) = \nabla\check{\varphi}_q\circ\nabla\check{\varphi}_p(\boldsymbol{\theta}_t) = W\cdot\bar{\boldsymbol{\theta}}_t$$

by Corollary B. We get

$$
\begin{aligned}
&D_{\varphi_q}\left(\bar{\boldsymbol{u}}\,\|\,\boldsymbol{w}_{t-1}\right) - D_{\varphi_q}\left(\bar{\boldsymbol{u}}\,\|\,\boldsymbol{w}_t\right)\\
&\ge (\bar{\boldsymbol{u}} - \boldsymbol{w}_{t-1})^\top\left(W\cdot\bar{\boldsymbol{\theta}}_t - \nabla\check{\varphi}_q(\boldsymbol{w}_{t-1})\right) - \frac{(p-1)W^2}{2}\cdot\left\|\bar{\boldsymbol{\theta}}_t - \frac{1}{W}\cdot\nabla\check{\varphi}_q\left(\boldsymbol{w}_{t-1}\right)\right\|_p^2\\
&= (\bar{\boldsymbol{u}} - \boldsymbol{w}_{t-1})^\top\left(W\cdot\bar{\boldsymbol{\theta}}_t - \nabla\check{\varphi}_q(\boldsymbol{w}_{t-1})\right) - \frac{p-1}{2}\cdot\left\|W\cdot\bar{\boldsymbol{\theta}}_t - \nabla\check{\varphi}_q\left(\boldsymbol{w}_{t-1}\right)\right\|_p^2\;.
\end{aligned}
$$

Now, note that

$$\boldsymbol{\theta}_t = \nabla\check{\varphi}_q(\boldsymbol{w}_{t-1}) + \eta_t\cdot(s_t - r_t)\cdot\boldsymbol{x}_t.$$

Figure 2: Schematic of proof of Lemma G. Arrows from equation $A$ to $B$ indicate that $A \implies B$.

We can thus rewrite the above as

$$
\begin{aligned}
D_{\varphi_q}\left(\bar{\boldsymbol{u}}\,\|\,\boldsymbol{w}_{t-1}\right) &- D_{\varphi_q}\left(\bar{\boldsymbol{u}}\,\|\,\boldsymbol{w}_t\right) \\
\geq\quad & s_t(s_t - r_t)\eta_t\alpha_t + (1-\alpha_t)\left(\boldsymbol{w}_{t-1}^\top \nabla\check\varphi_q(\boldsymbol{w}_{t-1}) - \bar{\boldsymbol{u}}^\top \nabla\check\varphi_q(\boldsymbol{w}_{t-1})\right) \\
& -\frac{p-1}{2}\cdot \left\|(s_t - r_t)\eta_t\alpha_t \cdot \boldsymbol{x}_t - (1-\alpha_t)\cdot \nabla\check\varphi_q(\boldsymbol{w}_{t-1})\right\|_p^2 \\
=\quad & s_t(s_t - r_t)\eta_t\alpha_t + (1-\alpha_t)\left(W^2 - \bar{\boldsymbol{u}}^\top \nabla\check\varphi_q(\boldsymbol{w}_{t-1})\right) \\
& -\frac{p-1}{2}\cdot \left\|(s_t - r_t)\eta_t\alpha_t \cdot \boldsymbol{x}_t - (1-\alpha_t)\cdot \nabla\check\varphi_q(\boldsymbol{w}_{t-1})\right\|_p^2 \text{ by definition of } \nabla\check\varphi_q \\
=\quad & Q + R + S + T \ ,
\end{aligned}
$$

as claimed. ∎

We can show that the sum $R + S + T \geq 0$. This proof involves chaining together multiple simple inequalities. We give a high level overview in Figure 2.

**Lemma G** *Let* $R, S, T$ *be as per Lemma F. Suppose we fix*

$$
\eta_t \;=\; \gamma \cdot \frac{W}{4(p-1)MX_pW + |y_t - \boldsymbol{w}_{t-1}^\top \boldsymbol{x}_t|X_p} \ , \tag{17}
$$

*for any* $\gamma \in [1/2, 1]$, *and* $M \doteq \max\{W, X_p\}$. *Then,* $T + R + S \geq 0$.

**Proof** The triangle inequality and the fact that $\|\nabla\check\varphi_q(\boldsymbol{w}_{t-1})\|_p = W$ brings

$$
\begin{aligned}
\alpha_t \;\in\; & \left[\frac{W}{\|\nabla\check\varphi_q(\boldsymbol{w}_{t-1})\|_p + \eta_t|s_t - r_t|\cdot \|\boldsymbol{x}_t\|_p}, \frac{W}{\|\nabla\check\varphi_q(\boldsymbol{w}_{t-1})\|_p - \eta_t|s_t - r_t|\cdot \|\boldsymbol{x}_t\|_p}\right] \\
\subseteq\; & \left[\frac{W}{W + \eta_t|s_t - r_t|X_p}, \frac{W}{W - \eta_t|s_t - r_t|\cdot X_p}\right] \ , \tag{18}
\end{aligned}
$$

*assuming* that $\eta_t$ is chosen so that

$$\eta_t \leq \frac{W}{|s_t - r_t| \cdot X_p} , \tag{19}$$

so that the right bound is non negative. To indeed ensure this, suppose that for some $0 < \epsilon_t \leq 1/2$, we fix

$$\eta_t \leq \frac{\epsilon_t}{1 - \epsilon_t} \cdot \frac{W}{|s_t - r_t|X_p} . \tag{20}$$

We would in addition obtain from Equation (18) that $\alpha_t \in [1 - \epsilon_t, 1 + \epsilon_t]$. Suppose $\eta_t$ is also fixed to ensure

$$\eta_t \in \left[ \frac{W}{2(p-1)\kappa_t X_p W^2 - |s_t - r_t|X_p}, \frac{W}{4(p-1)X_p W^2 + |s_t - r_t|X_p} \right] , \tag{21}$$

for some $\kappa_t$ such that

$$\kappa_t \geq 2 + \frac{|s_t - r_t|}{(p-1)W^2} . \tag{22}$$

Notice that constraint on $\kappa_t$ makes the interval non empty and its left bound strictly positive. Assuming (21) holds, we would have

$$\alpha_t \eta_t \in \left[ \frac{1}{2(p-1)\kappa_t X_p^2}, \frac{1}{4(p-1)X_p^2} \right] . \tag{23}$$

The left bound of (23) holds because

$$
\begin{aligned}
\alpha_t \eta_t &\geq \eta_t \cdot \frac{W}{W + \eta_t |s_t - r_t|X_p} \\
&\geq \frac{1}{2(p-1)\kappa_t X_p^2} .
\end{aligned}
\tag{24}
$$

The first inequality holds because of (18) and the second one holds because of (21). The right bound of (23) holds because of (18), and so

$$
\begin{aligned}
\alpha_t \eta_t &\leq \eta_t \cdot \frac{W}{W - \eta_t |s_t - r_t|X_p} \\
&\leq \frac{1}{4(p-1)X_p^2} ,
\end{aligned}
\tag{25}
$$

where the last inequality is due to (21). Equation (23) makes that $T(\eta_t \alpha_t)$ is at least its value when $\alpha_t \eta_t$ attains the lower bound of (24), that is,

$$T(\eta_t \alpha_t) \geq \frac{\kappa_t - 1}{\kappa_t} \cdot \frac{(s_t - r_t)^2}{8(p-1)X_p^2} . \tag{26}$$

Now, to guarantee $\alpha_t \in [1 - \epsilon_t, 1 + \epsilon_t]$, it is sufficient that the right-hand side of inequality (20) belongs to interval (21) *and* we pick $\eta_t$ within the interval [left bound (21), right-hand side (20)]. To guarantee that the right-hand side of inequality (20) falls in interval (21), we need first,

$$\frac{W}{2(p-1)\kappa_t X_p W^2 - |s_t - r_t| X_p} \leq \frac{\epsilon_t}{1 - \epsilon_t} \cdot \frac{W}{|s_t - r_t| X_p} \ , \tag{27}$$

that is,

$$\kappa_t \geq \frac{1}{\epsilon_t} \cdot \frac{|s_t - r_t|}{2(p-1)W^2} \ . \tag{28}$$

To guarantee that the right-hand side of inequality (20) falls in interval (21) we need then

$$\frac{W}{4(p-1)X_p W^2 + |s_t - r_t| X_p} \geq \frac{\epsilon_t}{1 - \epsilon_t} \cdot \frac{W}{|s_t - r_t| X_p} \ , \tag{29}$$

that is,

$$\epsilon_t \leq \frac{|s_t - r_t|}{2|s_t - r_t| + 4(p-1)W^2} \ . \tag{30}$$

To summarize, if we pick any strictly positive $\epsilon_t$ following inequality (30) (note $\epsilon_t < 1$) and

$$\kappa_t \doteq 2 + \frac{1}{\epsilon_t} \cdot \frac{|s_t - r_t|}{(p-1)W^2} \ , \tag{31}$$

then we shall have both $\alpha_t \in [1 - \epsilon_t, 1 + \epsilon_t]$ and inequality (26) holds as well. In this case, we shall have

$$
\begin{aligned}
T + R + S &\geq \left(1 - \frac{1}{2 + \frac{1}{\epsilon_t} \cdot \frac{|s_t - r_t|}{(p-1)W^2}}\right) \cdot \frac{(s_t - r_t)^2}{8(p-1)X_p^2} - 2\epsilon_t W^2 - (p-1)\epsilon_t^2 W^2 \\
&\geq \left(1 - \frac{1}{2}\right) \cdot \frac{(s_t - r_t)^2}{8(p-1)X_p^2} - 2\epsilon_t W^2 - (p-1)\epsilon_t^2 W^2 \\
&= \frac{(s_t - r_t)^2}{16(p-1)X_p^2} - 2\epsilon_t W^2 - (p-1)\epsilon_t^2 W^2 \ .
\end{aligned}
\tag{32}
$$

To finish up, we want to solve for $\epsilon_t$ the right-hand side such that it is non negative, and we find that $\epsilon_t$ has to satisfy

$$\epsilon_t \leq \frac{1}{p-1} \cdot \left(1 + \sqrt{1 + \frac{(s_t - r_t)^2}{16X_p^2 W^2}}\right) \ . \tag{33}$$

Since $\sqrt{1 + x} \geq \sqrt{x}$, a sufficient condition is

$$\epsilon_t \leq \frac{|s_t - r_t|}{4(p-1)X_p W} \ . \tag{34}$$

To ensure this and inequality (30), it is sufficient that we fix

$$\epsilon_t \doteq \frac{|s_t - r_t|}{2|s_t - r_t| + 4(p-1)WM} \ , \tag{35}$$

where $M \doteq \max\{W, X_p\}$. With this expression for $\epsilon_t$, we get from (31),

$$\kappa_t \doteq 2 + \frac{4M}{W} + \frac{2|s_t - r_t|}{(p-1)W^2} \ . \tag{36}$$

For these choices, Lemma H implies that the given $\eta_t$ is feasible. ∎

**Lemma H** *Suppose $\epsilon_t$ satisfies (35) and $\kappa_t$ satisfies (36). Then, a sufficient condition for $\eta_t$ to satisfy both (20) and (21) is*

$$\eta_t = \gamma \cdot \frac{W}{4(p-1)MX_pW + |y_t - \boldsymbol{w}_{t-1}^\top \boldsymbol{x}_t|X_p} \ ,$$

*for any $\gamma \in [1/2, 1]$.*

**Proof** [Proof of Lemma H] Notice the range of values authorized for $\eta_t$:

$$\eta_t \in \left[ \frac{W}{2(p-1)\kappa_t X_p W^2 - |s_t - r_t|X_p}, \frac{\epsilon_t}{1 - \epsilon_t} \cdot \frac{W}{|s_t - r_t|X_p} \right]$$

$$= \left[ \frac{W}{2(p-1)\left(2 + \frac{4M}{W} + \frac{2|s_t - r_t|}{(p-1)W^2}\right)X_p W^2 - |s_t - r_t|X_p}, \frac{W}{4(p-1)MX_pW + |s_t - r_t|X_p} \right]$$

$$= \left[ \frac{W}{2(2(p-1)W^2 + 4M(p-1)W + 2|s_t - r_t|)X_p - |s_t - r_t|X_p}, \frac{W}{4(p-1)MX_pW + |s_t - r_t|X_p} \right]$$

$$= \left[ \frac{W}{4(p-1)X_pW^2 + 8(p-1)MX_pW + 3|s_t - r_t|X_p}, \frac{W}{4(p-1)MX_pW + |s_t - r_t|X_p} \right]$$

$$\supset \left[ \frac{W}{8(p-1)MX_pW + 2|s_t - r_t|X_p}, \frac{W}{4(p-1)MX_pW + |s_t - r_t|X_p} \right] \ . \tag{37}$$

A sufficient condition for $\eta_t$ to fall in interval (37) is

$$\eta_t = \gamma \cdot \frac{W}{4(p-1)MX_pW + |y_t - \boldsymbol{w}_{t-1}^\top \boldsymbol{x}_t|X_p} \ ,$$

for any $\gamma \in [1/2, 1]$. ∎

**Lemma I** *Suppose we fix the learning rate as per (17). Pick any $\boldsymbol{u}$, and consider iterates $\{\boldsymbol{w}_t\}_{t=0}^T$ as per Equation 10. Suppose $\|\boldsymbol{x}_t\|_p \leq X_p$ and $|y_t| \leq Y, \forall t \leq T$. Then, for any $t$,*

$$D_{\varphi_q}\left(\bar{\boldsymbol{u}} \,\|\, \boldsymbol{w}_{t-1}\right) - D_{\varphi_q}\left(\bar{\boldsymbol{u}} \,\|\, \boldsymbol{w}_t\right) \geq \frac{1}{4(p-1)\left(2 + \frac{4M}{W} + \frac{2(Y+X_pW)}{(p-1)W^2}\right) X_p^2} \cdot (s_t^2 - r_t^2)$$

*where $s_t \doteq (\bar{\boldsymbol{u}} - \boldsymbol{w}_{t-1})^\top \boldsymbol{x}_t$, $r_t \doteq \bar{\boldsymbol{u}}^\top \boldsymbol{x}_t - y_t$.*

**Proof** [Proof of Lemma I] We start from the bound of Lemma F:

$$
\begin{aligned}
D_{\varphi_q}\left(\bar{\boldsymbol{u}} \,\|\, \boldsymbol{w}_{t-1}\right) - D_{\varphi_q}\left(\bar{\boldsymbol{u}} \,\|\, \boldsymbol{w}_t\right) &\geq Q + R + S + T \\
&\geq Q \text{ by Lemma G} \\
&= \frac{\alpha_t}{2}\eta_t(s_t^2 - r_t^2) \text{ by definition} \\
&\geq \frac{1}{4(p-1)\kappa_t X_p^2} \cdot (s_t^2 - r_t^2) \\
&\geq \frac{1}{4(p-1)\left(2 + \frac{4M}{W} + \frac{2\max_t |y_t - \boldsymbol{w}_{t-1}^\top \boldsymbol{x}_t|}{(p-1)W^2}\right) X_p^2} \cdot (s_t^2 - r_t^2) \\
&\geq \frac{1}{4(p-1)\left(2 + \frac{4M}{W} + \frac{2(Y+X_pW)}{(p-1)W^2}\right) X_p^2} \cdot (s_t^2 - r_t^2) \ . \qquad (38)
\end{aligned}
$$

The last constraint to check for this bound to be valid is our $\epsilon_t$ in (35) has to be $< 1/2$ from inequality (19), which trivially holds since $4(p-1)WM \geq 0$. We conclude by noting Lemma H provides a feasible value of $\eta_t$. ∎

## II.4   Gauge normalisation

The following lemma about the gauge of $\boldsymbol{x}$ will be useful.

**Lemma J** *Let $g_q(\boldsymbol{x}) = \|\boldsymbol{x}\|_q/W$ for some $W > 0$. Then, for the iterates $\{\boldsymbol{w}_t\}$ as per Equation 9, $g_q(\boldsymbol{w}_t) = 1$.*

**Proof** We have

$$
\begin{aligned}
g_q(\boldsymbol{w}_t) &= \frac{\|\boldsymbol{w}_t\|_q}{W} \\
&= \frac{W}{W} \text{ by Lemma 3} \\
&= 1 \ , \forall t \geq 1 \ . \qquad (39)
\end{aligned}
$$

∎

# III  Working out examples of Table A1

We fill in the details justifying each of the examples of Equation 3 provided in Table 2. We also provide the form of the corresponding divergences $D_\varphi$ and distortions $D_{\check\varphi}$ in the augmented Table A1.

| | $\varphi$ | $D_\varphi\left(\boldsymbol{x}\|\boldsymbol{y}\right)$ | $g$ | $\check\varphi$ | $D_{\check\varphi}\left(\boldsymbol{x}\|\boldsymbol{y}\right)$ |
|---|---|---|---|---|---|
| I | $\frac{1}{2}\cdot(1+\|\boldsymbol{x}\|_2^2)$ | $(1/2)\cdot\|\boldsymbol{x}-\boldsymbol{y}\|_2^2$ | $\|\boldsymbol{x}\|_2$ | $\|\boldsymbol{x}\|_2$ | $\|\boldsymbol{x}\|_2\cdot(1-\cos\angle\boldsymbol{x},\boldsymbol{y})$ |
| II | $\frac{1}{2}\cdot(W+\|\boldsymbol{x}\|_q^2)$ | $(1/2)\cdot(\|\boldsymbol{x}\|_q^2-\|\boldsymbol{y}\|_q^2)$ $-\sum_i\frac{(x_i-y_i)\cdot\text{sign}(y_i)\cdot|y_i|^{q-1}}{\|\boldsymbol{y}\|_q^{q-2}}$ | $\frac{\|\boldsymbol{x}\|_q}{W}$ | $W\cdot\|\boldsymbol{x}\|_q$ | $W\cdot|\boldsymbol{x}|_q-W\cdot\sum_i\frac{x_i\cdot\text{sign}(y_i)\cdot|y_i|^{q-1}}{\|\boldsymbol{y}\|_q^{q-1}}$ |
| III | $\frac{1}{2}\cdot(u^2+\|\boldsymbol{x}^S\|_2^2)$ | $(1/2)\cdot\|\boldsymbol{x}^S-\boldsymbol{y}^S\|_2^2$ | $\frac{\|\boldsymbol{x}\|_2}{\sin\|\boldsymbol{x}\|_2}$ | $\|\boldsymbol{x}^S\|_2$ | $\frac{\|\boldsymbol{x}\|_2}{\sin\|\boldsymbol{x}\|_2}\cdot(1-\cos D_G(\boldsymbol{x},\boldsymbol{y}))$ |
| IV | $\frac{1}{2}\cdot(u^2+\|\boldsymbol{x}^H\|_2^2)$ | $(1/2)\cdot\|\boldsymbol{x}^H-\boldsymbol{y}^H\|_2^2$ | $-\frac{\|\boldsymbol{x}\|_2}{\sinh\|\boldsymbol{x}\|_2}$ | $\|\boldsymbol{x}^H\|_2$ | $-\frac{\|\boldsymbol{x}\|_2}{\sinh\|\boldsymbol{x}\|_2}\cdot(\cosh D_G(\boldsymbol{x},\boldsymbol{y})-1)$ |
| V | $\sum_i x_i\log x_i-x_i$ | $\sum_i x_i\log\frac{x_i}{y_i}$ $-\mathbf{1}^\top(\boldsymbol{x}-\boldsymbol{y})$ | $\mathbf{1}^\top\boldsymbol{x}$ | $\sum_i x_i\log x_i-\mathbf{1}^\top\boldsymbol{x}$ $-(\mathbf{1}^\top\boldsymbol{x})\log(\mathbf{1}^\top\boldsymbol{x})$ | $\sum_i x_i\log\frac{x_i}{y_i}$ $-d\cdot\mathbb{E}[\mathsf{X}]\cdot\log\frac{\mathbb{E}[\mathsf{X}]}{\mathbb{E}[\mathsf{Y}]}$ |
| VI | $-d-\sum_i\log x_i$ | $\sum_i\frac{x_i}{y_i}$ $-\sum_i\log\frac{x_i}{y_i}-d$ | $\prod_i x_i^{1/d}$ | $-d\cdot\prod_i x_i^{1/d}$ | $\sum_i\frac{x_i(\pi_{\boldsymbol{y}})^{1/d}}{y_i}-d(\pi_{\boldsymbol{x}})^{1/d}$ |
| VII | $\text{tr}\,(\mathsf{x}\log\mathsf{x}-\mathsf{x})$ | $\text{tr}\,(\mathsf{x}\log\mathsf{x}-\mathsf{x}\log\mathsf{Y})$ $-\text{tr}\,(\mathsf{x})+\text{tr}\,(\mathsf{Y})$ | $\text{tr}\,(\mathsf{x})$ | $\text{tr}\,(\mathsf{x}\log\mathsf{x}-\mathsf{x})$ $-\text{tr}\,(\mathsf{x})\log\text{tr}\,(\mathsf{x})$ | $\text{tr}\,(\mathsf{x}\log\mathsf{x}-\mathsf{x}\log\mathsf{Y})$ $-\text{tr}\,(\mathsf{x})\cdot\log\frac{\text{tr}(\mathsf{X})}{\text{tr}(\mathsf{Y})}$ |
| VIII | $-d-\log\det(\mathsf{x})$ | $\text{tr}\,(\mathsf{x}\mathsf{Y}^{-1})$ $-\log\det(\mathsf{x}\mathsf{Y}^{-1})-d$ | $\det(\mathsf{x}^{1/d})$ | $-d\cdot\det(\mathsf{x}^{1/d})$ | $\det(\mathsf{Y}^{1/d})\text{tr}\,(\mathsf{x}\mathsf{Y}^{-1})-d\cdot\det(\mathsf{x}^{1/d})$ |

Table A1: Example of distortions (right columns) that can be "reverse engineered" as Bregman divergences involving a particular, non necessary linear $g$. Function $\boldsymbol{x}^S\doteq f(\boldsymbol{x}):\mathbb{R}^d\to\mathbb{R}^{d+1}$ is the (S)phere lifting map defined in (51), and $\boldsymbol{x}^H$ is the (H)yperboloid lifting map defined in (62). $D_G(.,.)$ is the geodesic distance between the exponential map of $\boldsymbol{x}$ and $\boldsymbol{y}$ on their respective manifold (sphere or hyperboloid). Related proofs are in Section III. Expectation $\mathbb{E}[\mathsf{X}]$ is a shorthand for $(1/d)\cdot\sum_i x_i$. $W\in\mathbb{R}_{+*}$ and $u\in\mathbb{R}$ are constants.

**Row I —**  for $\mathcal{X}=\mathbb{R}^d$, consider $\varphi(\boldsymbol{x})=(1+\|\boldsymbol{x}\|_2^2)/2$ and $g(\boldsymbol{x})=\|\boldsymbol{x}\|_2$ (we project on the Euclidean sphere). It comes

$$\check\varphi(\boldsymbol{x}) \;=\; \|\boldsymbol{x}\|_2\cdot\left(\frac{1+\left\|\frac{1}{\|\boldsymbol{x}\|_2}\cdot\boldsymbol{x}\right\|_2^2}{2}\right)=\|\boldsymbol{x}\|_2\ . \tag{40}$$

$g$ is not linear (but it is homogeneous of degree 1), but we have

$$\varphi(\boldsymbol{x})=1=\boldsymbol{x}^\top\nabla\varphi(\boldsymbol{x})\ ,\forall\boldsymbol{x}:\|\boldsymbol{x}\|_2=1\ , \tag{41}$$

so $\varphi$ is 1-homogeneous on the Euclidean sphere, and we can apply Theorem 1. We have

$$
\begin{aligned}
g(\boldsymbol{x}) \cdot D_\varphi \left( \frac{1}{g(\boldsymbol{x})} \cdot \boldsymbol{x} \Big\| \frac{1}{g(\boldsymbol{y})} \cdot \boldsymbol{y} \right) &= \frac{\|\boldsymbol{x}\|_2}{2} \cdot \left\| \frac{1}{\|\boldsymbol{x}\|_2} \cdot \boldsymbol{x} - \frac{1}{\|\boldsymbol{y}\|_2} \cdot \boldsymbol{y} \right\|_2^2 \\
&= \|\boldsymbol{x}\|_2 \cdot \left( 1 - \frac{\boldsymbol{x}^\top \boldsymbol{y}}{\|\boldsymbol{x}\|_2 \|\boldsymbol{y}\|_2} \right) = \|\boldsymbol{x}\|_2 \cdot (1 - \cos(\boldsymbol{x}, \boldsymbol{y})) \ , (42)
\end{aligned}
$$

and we also have

$$
\begin{aligned}
D_{\check{\varphi}}(\boldsymbol{x}\|\boldsymbol{y}) &= \|\boldsymbol{x}\|_2 - \|\boldsymbol{y}\|_2 - \frac{1}{\|\boldsymbol{y}\|_2} \cdot (\boldsymbol{x} - \boldsymbol{y})^\top \boldsymbol{y} \\
&= \|\boldsymbol{x}\|_2 - \|\boldsymbol{y}\|_2 - \frac{\boldsymbol{x}^\top \boldsymbol{y}}{\|\boldsymbol{y}\|_2} + \|\boldsymbol{y}\|_2 \tag{43} \\
&= \|\boldsymbol{x}\|_2 \cdot \left( 1 - \frac{\boldsymbol{x}^\top \boldsymbol{y}}{\|\boldsymbol{x}\|_2 \|\boldsymbol{y}\|_2} \right) = \|\boldsymbol{x}\|_2 \cdot (1 - \cos(\boldsymbol{x}, \boldsymbol{y})) \ , \tag{44}
\end{aligned}
$$

which is equal to Equation (42), so we check that Theorem 1 applies in this case. $D_{\check{\varphi}}$ has some interesting properties. One is a weak form of triangle inequality.

**Lemma K** $D_{\check{\varphi}}(\boldsymbol{x}\|\boldsymbol{y}) + D_{\check{\varphi}}(\boldsymbol{y}\|\boldsymbol{z}) \leq D_{\check{\varphi}}(\boldsymbol{x}\|\boldsymbol{z})$, $\forall \boldsymbol{x}, \boldsymbol{y}, \boldsymbol{z}$ such that $\|\boldsymbol{y}\|_2 \leq \|\boldsymbol{x}\|_2$.

**Proof**

$$
\begin{aligned}
D_{\check{\varphi}}&(\boldsymbol{x}\|\boldsymbol{y}) + D_{\check{\varphi}}(\boldsymbol{y}\|\boldsymbol{z}) \\
&= \|\boldsymbol{x}\|_2 \cdot (1 - \cos(\boldsymbol{x}, \boldsymbol{y})) + \|\boldsymbol{y}\|_2 \cdot (1 - \cos(\boldsymbol{y}, \boldsymbol{z})) \\
&= \|\boldsymbol{x}\|_2 \cdot ((1 - \cos(\boldsymbol{x}, \boldsymbol{y})) + (1 - \cos(\boldsymbol{y}, \boldsymbol{z}))) + (\|\boldsymbol{y}\|_2 - \|\boldsymbol{x}\|_2) \cdot (1 - \cos(\boldsymbol{y}, \boldsymbol{z})) \\
&\leq \|\boldsymbol{x}\|_2 \cdot (1 - \cos(\boldsymbol{x}, \boldsymbol{z})) + (\|\boldsymbol{y}\|_2 - \|\boldsymbol{x}\|_2) \cdot (1 - \cos(\boldsymbol{y}, \boldsymbol{z})) \\
&\leq D_{\check{\varphi}}(\boldsymbol{x}\|\boldsymbol{z}) + (\|\boldsymbol{y}\|_2 - \|\boldsymbol{x}\|_2) \cdot (1 - \cos(\boldsymbol{y}, \boldsymbol{z})) \\
&\leq D_{\check{\varphi}}(\boldsymbol{x}\|\boldsymbol{z}) \ , \tag{45}
\end{aligned}
$$

since $\|\boldsymbol{y}\|_2 \leq \|\boldsymbol{x}\|_2$. We have used the fact that $(1 - \cos(\boldsymbol{x}, \boldsymbol{y}))$ is half the Euclidean distance between unit-normalized vectors. ∎

Another good property is that $D_{\check{\varphi}}(\boldsymbol{x}\|\boldsymbol{\mu})$ can be related to the log-likelihood of a von Mises-Fisher distribution with expected direction $\boldsymbol{\mu}$, which happens to be useful in text analysis [Reisinger et al., 2010].

**Row II —** Let $\varphi(\boldsymbol{x}) \doteq (1/2) \cdot (u^2 + \|\boldsymbol{x}\|_q^2)$, for $q > 1$ [Kivinen et al., 2006]. We have

$$
\varphi\left( \frac{1}{g(\boldsymbol{x})} \cdot \boldsymbol{x} \right) = \frac{u^2}{2} + \frac{1}{2} \cdot \left\| \frac{1}{g(\boldsymbol{x})} \cdot \boldsymbol{x} \right\|_q^2 = \frac{u^2}{2} + \frac{1}{2g^2(\boldsymbol{x})} \cdot \|\boldsymbol{x}\|_q^2 \ . \tag{46}
$$

We also have

$$\left(\frac{1}{g(\boldsymbol{x})}\cdot\boldsymbol{x}\right)^{\top}\nabla\varphi\left(\frac{1}{g(\boldsymbol{x})}\cdot\boldsymbol{x}\right) \;=\; \frac{1}{g(\boldsymbol{x})}\cdot\sum_{i}\frac{x_i\cdot\mathrm{sign}\left(\frac{1}{g(\boldsymbol{x})}\cdot x_i\right)\left|\frac{1}{g(\boldsymbol{x})}\cdot x_i\right|^{q-1}}{\left\|\frac{1}{g(\boldsymbol{x})}\cdot\boldsymbol{x}\right\|_q^{q-2}}$$

$$=\; \sum_{i}\frac{\left|\frac{1}{g(\boldsymbol{x})}\cdot x_i\right|^{q}}{\left\|\frac{1}{g(\boldsymbol{x})}\cdot\boldsymbol{x}\right\|_q^{q-2}}$$

$$=\; \frac{1}{g^2(\boldsymbol{x})}\cdot\|\boldsymbol{x}\|_q^2\;. \tag{47}$$

To have the condition of Theorem 1 satisfied, we therefore need

$$\|\boldsymbol{x}\|_q \;=\; ug(\boldsymbol{x})\;, \tag{48}$$

So we use $g(\boldsymbol{x})=\|\boldsymbol{x}\|_q/W$ and $u=W$, observing that $\varphi$ is 1-homogeneous on the $L_p$ sphere. We check that

$$\check{\varphi}(\boldsymbol{x}) \;=\; W\cdot\|\boldsymbol{x}\|_q\;. \tag{49}$$

and we obtain

$$D_{\check{\varphi}}(\boldsymbol{w}\|\boldsymbol{w}') \;=\; W\cdot\|\boldsymbol{w}\|_q - W\cdot\sum_{i}\frac{w_i\cdot\mathrm{sign}(w_i')\cdot|w_i'|^{q-1}}{\|\boldsymbol{w}'\|_q^{q-1}}\;. \tag{50}$$

**Row III —** As in Buss and Fillmore [2001], we assume $\|\boldsymbol{x}\|_2 \leq \pi$, or we renormalize or change the radius of the ball) We first lift the data points using the *Sphere* lifting map $\mathbb{R}^d \ni \boldsymbol{x} \mapsto \boldsymbol{x}^S \in \mathbb{R}^{d+1}$:

$$\boldsymbol{x}^S \;\doteq\; [x_1 \quad x_2 \quad \cdots \quad x_d \quad r_{\boldsymbol{x}}\cot r_{\boldsymbol{x}}]^{\top}\;, \tag{51}$$

where $r_{\boldsymbol{x}} \doteq \|\boldsymbol{x}\|_2$ is the Euclidean norm of $\boldsymbol{x}$. Notice that the last coordinate is a coordinate of the Hessian of the geodesic distance to the origin on the sphere [Buss and Fillmore, 2001]. We then let $g(\boldsymbol{x}^S) \doteq r_{\boldsymbol{x}}/\sin r_{\boldsymbol{x}}$ (notice that $g$ is computed uding the first $d$ coordinates). Finally, for $\mathcal{X}=\mathbb{R}^{d+1}$ and $u>1$, consider $\varphi(\boldsymbol{x}^S)=(u^2+\|\boldsymbol{x}^S\|_2^2)/2$. The set of points for which $\varphi(\boldsymbol{x}^S)=(\boldsymbol{x}^S)^{\top}\nabla\varphi(\boldsymbol{x}^S)$ is equivalently the subset $\mathcal{X}_g \subseteq \mathbb{R}^{d+1}$ such that

$$\mathcal{X}_g \;\doteq\; \{\boldsymbol{x}^S : g^2(\boldsymbol{x}^S)=u^2\}\;. \tag{52}$$

So $\varphi$ satisfies the restricted positive homogeneity of degree 1 on $\mathcal{X}_g$ and we can apply Theorem 1. We first remark that:

$$\|\boldsymbol{x}^S\|_2^2 \;=\; r_{\boldsymbol{x}}^2 + r_{\boldsymbol{x}}^2\cot^2 r_{\boldsymbol{x}}$$

$$=\; \frac{r_{\boldsymbol{x}}^2}{\sin r_{\boldsymbol{x}}^2} = g^2(\boldsymbol{x}^S)\;, \tag{53}$$

and

$$\check{\varphi}(\boldsymbol{x}^S) \;=\; \frac{r_{\boldsymbol{x}}}{\sin r_{\boldsymbol{x}}} \cdot \varphi\left(\frac{\sin r_{\boldsymbol{x}}}{r_{\boldsymbol{x}}} \cdot \boldsymbol{x}^S\right) = \frac{r_{\boldsymbol{x}}}{\sin r_{\boldsymbol{x}}} \cdot \left(\frac{\sin r_{\boldsymbol{x}}}{r_{\boldsymbol{x}}}\right)^2 \cdot \|\boldsymbol{x}^S\|_2^2 = \|\boldsymbol{x}^S\|_2 \;, \tag{54}$$

and finally, because of the spherical law of cosines,

$$\sin r_{\boldsymbol{x}} \sin r_{\boldsymbol{y}} \cos(\boldsymbol{x}, \boldsymbol{y}) + \cos r_{\boldsymbol{x}} \cos r_{\boldsymbol{y}} \;=\; \cos D_G(\boldsymbol{x}, \boldsymbol{y}) \;, \tag{55}$$

where we recall from eq. (16) that $D_G(\boldsymbol{x}, \boldsymbol{y})$ is the geodesic distance between the image of the exponential maps of $\boldsymbol{x}$ and $\boldsymbol{y}$ on the sphere. We then derive

$$g(\boldsymbol{x}^S) \cdot D_\varphi\left(\frac{1}{g(\boldsymbol{x}^S)} \cdot \boldsymbol{x}^S \| \frac{1}{g(\boldsymbol{y}^S)} \cdot \boldsymbol{y}^S\right)$$

$$= \frac{r_{\boldsymbol{x}}}{2\sin r_{\boldsymbol{x}}} \cdot \left\|\frac{\sin r_{\boldsymbol{x}}}{r_{\boldsymbol{x}}} \cdot \boldsymbol{x}^S - \frac{\sin r_{\boldsymbol{y}}}{r_{\boldsymbol{y}}} \cdot \boldsymbol{y}^S\right\|_2^2$$

$$= \frac{r_{\boldsymbol{x}}}{2\sin r_{\boldsymbol{x}}} \cdot \left(\frac{\sin^2 r_{\boldsymbol{x}}}{\|\boldsymbol{x}\|_2^2} \cdot \|\boldsymbol{x}^S\|_2^2 + \frac{\sin^2 r_{\boldsymbol{y}}}{\|\boldsymbol{y}\|_2^2} \cdot \|\boldsymbol{y}^S\|_2^2 - 2 \cdot \frac{\sin r_{\boldsymbol{x}}}{r_{\boldsymbol{x}}} \cdot \frac{\sin r_{\boldsymbol{y}}}{r_{\boldsymbol{y}}} \cdot (\boldsymbol{x}^S)^\top \boldsymbol{y}^S\right)$$

$$= \frac{r_{\boldsymbol{x}}}{\sin r_{\boldsymbol{x}}} \cdot \left(1 - \frac{\sin r_{\boldsymbol{x}}}{r_{\boldsymbol{x}}} \cdot \frac{\sin r_{\boldsymbol{y}}}{r_{\boldsymbol{y}}} \cdot (\boldsymbol{x}^S)^\top \boldsymbol{y}^S\right) \tag{56}$$

$$= \frac{r_{\boldsymbol{x}}}{\sin r_{\boldsymbol{x}}} \cdot \left(1 - \frac{\sin r_{\boldsymbol{x}}}{r_{\boldsymbol{x}}} \cdot \frac{\sin r_{\boldsymbol{y}}}{r_{\boldsymbol{y}}} \cdot \left(\boldsymbol{x}^\top \boldsymbol{y} + r_{\boldsymbol{x}} r_{\boldsymbol{y}} \cot r_{\boldsymbol{x}} \cot r_{\boldsymbol{y}}\right)\right) \tag{57}$$

$$= \frac{r_{\boldsymbol{x}}}{\sin r_{\boldsymbol{x}}} \cdot \left(1 - \sin r_{\boldsymbol{x}} \sin r_{\boldsymbol{y}} \cdot (\cos(\boldsymbol{x}, \boldsymbol{y}) + \cot r_{\boldsymbol{x}} \cot r_{\boldsymbol{y}})\right)$$

$$= \frac{r_{\boldsymbol{x}}}{\sin r_{\boldsymbol{x}}} \cdot \left(1 - (\sin r_{\boldsymbol{x}} \sin r_{\boldsymbol{y}} \cos(\boldsymbol{x}, \boldsymbol{y}) + \cos r_{\boldsymbol{x}} \cos r_{\boldsymbol{y}})\right)$$

$$= \frac{r_{\boldsymbol{x}}}{\sin r_{\boldsymbol{x}}} \cdot (1 - \cos D_G(\boldsymbol{x}, \boldsymbol{y})) \;. \tag{58}$$

In Equation (56), we use Equation (53), and we use Equation (55) in Equation (58). We also check

$$D_{\check{\varphi}}(\boldsymbol{x}^S \| \boldsymbol{y}^S) \;=\; \|\boldsymbol{x}^S\|_2 - \|\boldsymbol{y}^S\|_2 - \frac{1}{\|\boldsymbol{y}^S\|_2} \cdot (\boldsymbol{x}^S - \boldsymbol{y}^S)^\top \boldsymbol{y}^S$$

$$= \|\boldsymbol{x}^S\|_2 - \frac{1}{\|\boldsymbol{y}^S\|_2} \cdot (\boldsymbol{x}^S)^\top \boldsymbol{y}^S$$

$$= \|\boldsymbol{x}^S\|_2 \cdot \left(1 - \frac{(\boldsymbol{x}^S)^\top \boldsymbol{y}^S}{\|\boldsymbol{x}^S\|_2 \|\boldsymbol{y}^S\|_2}\right) \tag{59}$$

$$= \frac{r_{\boldsymbol{x}}}{\sin r_{\boldsymbol{x}}} \cdot (1 - \cos D_G(\boldsymbol{x}, \boldsymbol{y})) \;. \tag{60}$$

To obtain (60), we use the fact that

$$\frac{(\boldsymbol{x}^S)^\top \boldsymbol{y}^S}{\|\boldsymbol{x}^S\|_2 \|\boldsymbol{y}^S\|_2} \;=\; \frac{\sin r_{\boldsymbol{x}}}{r_{\boldsymbol{x}}} \cdot \frac{\sin r_{\boldsymbol{y}}}{r_{\boldsymbol{y}}} \cdot \left(\boldsymbol{x}^\top \boldsymbol{y} + r_{\boldsymbol{x}} r_{\boldsymbol{y}} \cot r_{\boldsymbol{x}} \cot r_{\boldsymbol{y}}\right) \;, \tag{61}$$

and then plug it into Equation (60), which yields the identity between Equation (57) (and thus (58)) and (60). So Theorem 1 holds in this case as well. We also remark that $(1/g(\boldsymbol{x}^S)) \cdot \boldsymbol{x}^S = \exp_{\boldsymbol{0}}(\boldsymbol{x})$ is the exponential map for the sphere [Buss and Fillmore, 2001].

**Row IV** — In the same way as we did for row IV, we first create a lifting map, but this time *complex* valued, the Hyperboloid lifting map $H\colon \mathbb{R}^d \ni \boldsymbol{x} \mapsto \boldsymbol{x}^H \in \mathbb{R}^d \times \mathbb{C}$. With an abuse of notation, it is given by

$$\boldsymbol{x}^H \;\dot{=}\; [x_1 \quad x_2 \quad \cdots \quad x_d \quad i r_{\boldsymbol{x}} \coth r_{\boldsymbol{x}}]^\top \;, \tag{62}$$

and we let $g(\boldsymbol{x}^H) \doteq -r_{\boldsymbol{x}}/\sinh r_{\boldsymbol{x}}$, with $\coth$ and $\sinh$ defining respectively the hyperbolic cotangent and hyperbolic sine. We let $0 \coth 0 = 0/\sinh 0 = 1$. Notice that the complex number is pure imaginary and so $H$ defines a $d$ dimensional manifold that lives in $\mathbb{R}^{d+1}$ assuming that the last coordinate is the imaginary axis. Let $\exp_{\boldsymbol{q}}(\boldsymbol{x}) \doteq (1/g(\boldsymbol{x}^H)) \cdot \boldsymbol{x}^H$. Notice that

$$
\begin{aligned}
\left\| \exp_{\boldsymbol{q}}(\boldsymbol{x}) \right\|_2^2 
&= \frac{\sinh^2 r_{\boldsymbol{x}}}{r_{\boldsymbol{x}}^2} \cdot \left( r_{\boldsymbol{x}}^2 + i^2 r_{\boldsymbol{x}}^2 \coth^2 r_{\boldsymbol{x}} \right) \\
&= \sinh^2 r_{\boldsymbol{x}} + i^2 \cosh^2 r_{\boldsymbol{x}} \\
&= \sinh^2 r_{\boldsymbol{x}} - \cosh^2 r_{\boldsymbol{x}} = -1 \;,
\end{aligned}
\tag{63}
$$

so $\exp_{\boldsymbol{q}}(\boldsymbol{x})$ defines a lifting map from $\mathbb{R}^d$ to the hyperboloid model $\mathbb{H}^d$ of hyperbolic geomety [Galperin, 1993]. In fact, it defines the exponential map for the plane $T_{\boldsymbol{q}}\mathbb{H}^d$ tangent to $\mathbb{H}^d$ in point

$$\boldsymbol{q} \doteq [0 \quad 0 \quad \cdots \quad 0 \quad i] = \boldsymbol{0}^H.$$

To see this, remark that we can express the geodesic distance $D_G$ with the hyperbolic metric between $\boldsymbol{x}^H$ and $\boldsymbol{y}^H$ as

$$D_G(\boldsymbol{x}^H, \boldsymbol{y}^H) \;\dot{=}\; \cosh^{-1}(-(\boldsymbol{x}^H)^\top \boldsymbol{y}^H) \;, \tag{64}$$

where $\cosh^{-1}$ is the inverse hyperbolic cosine. So, for any $\boldsymbol{x} \in T_{\boldsymbol{q}}\mathbb{H}^d$, since $r_{\boldsymbol{x}} = \|\boldsymbol{x} - \boldsymbol{0}\|_2$, we have

$$
\begin{aligned}
D_G(\exp_{\boldsymbol{q}}(\boldsymbol{x}), \boldsymbol{q}) 
&= \cosh^{-1}(-(\boldsymbol{x}^H)^\top \boldsymbol{0}^H) \\
&= \cosh^{-1}(-i^2 \cosh r_{\boldsymbol{x}}) \\
&= r_{\boldsymbol{x}} = \|\boldsymbol{x} - \boldsymbol{0}\|_2 \;,
\end{aligned}
\tag{65}
$$

and $\exp_{\boldsymbol{q}}(\boldsymbol{x})$ is indeed the exponential map for $T_{\boldsymbol{q}}\mathbb{H}^d$. Now, remark that

$$
\begin{aligned}
\exp_{\boldsymbol{q}}(\boldsymbol{x})^\top \exp_{\boldsymbol{q}}(\boldsymbol{y}) 
&= \frac{\sinh r_{\boldsymbol{x}}}{r_{\boldsymbol{x}}} \cdot \frac{\sinh r_{\boldsymbol{y}}}{r_{\boldsymbol{y}}} \cdot (\boldsymbol{x}^H)^\top \boldsymbol{y}^H \\
&= \frac{\sinh r_{\boldsymbol{x}}}{r_{\boldsymbol{x}}} \cdot \frac{\sinh r_{\boldsymbol{y}}}{r_{\boldsymbol{y}}} \cdot \left( \boldsymbol{x}^\top \boldsymbol{y} + i^2 r_{\boldsymbol{x}} r_{\boldsymbol{y}} \coth r_{\boldsymbol{x}} \coth r_{\boldsymbol{y}} \right) \\
&= \sinh r_{\boldsymbol{x}} \sinh r_{\boldsymbol{y}} \cdot (\cos(\boldsymbol{x}, \boldsymbol{y}) - \coth r_{\boldsymbol{x}} \coth r_{\boldsymbol{y}}) \\
&= \sinh r_{\boldsymbol{x}} \sinh r_{\boldsymbol{y}} \cos(\boldsymbol{x}, \boldsymbol{y}) - \cosh r_{\boldsymbol{x}} \cosh r_{\boldsymbol{y}} \\
&= -\cosh D_G(\boldsymbol{x}^H, \boldsymbol{y}^H) \;.
\end{aligned}
\tag{66}
$$

Eq. (66) holds by the hyperbolic law of cosines. Now, we let $\varphi(\boldsymbol{x}^H) = (u^2 + \|\boldsymbol{x}^H\|_2^2)/2$ and

$$\mathcal{X}_g \;\dot{=}\; \{\boldsymbol{x}^H : \|\boldsymbol{x}^H\|_2^2 = u^2\} \;. \tag{67}$$

We check that $\varphi(\boldsymbol{x}^H) = u^2 = (\boldsymbol{x}^H)^\top \nabla \varphi(\boldsymbol{x}^H)$ for any $\boldsymbol{x}^H \in \mathfrak{X}_g$, so we can apply Theorem 1. We then use eqs. (63) and (66) and derive

$$
\begin{aligned}
g(\boldsymbol{x}^H) \cdot D_\varphi &\left( \frac{1}{g(\boldsymbol{x}^H)} \cdot \boldsymbol{x}^H \middle\| \frac{1}{g(\boldsymbol{y}^H)} \cdot \boldsymbol{y}^H \right) \\
&= -\frac{r_{\boldsymbol{x}}}{2 \sinh r_{\boldsymbol{x}}} \cdot \left\| \exp_{\boldsymbol{q}}(\boldsymbol{x}) - \exp_{\boldsymbol{q}}(\boldsymbol{y}) \right\|_2^2 \\
&= -\frac{r_{\boldsymbol{x}}}{2 \sinh r_{\boldsymbol{x}}} \cdot \left( \left\| \exp_{\boldsymbol{q}}(\boldsymbol{x}) \right\|_2^2 + \left\| \exp_{\boldsymbol{q}}(\boldsymbol{y}) \right\|_2^2 - 2 \exp_{\boldsymbol{q}}(\boldsymbol{x})^\top \exp_{\boldsymbol{q}}(\boldsymbol{y}) \right) \\
&= -\frac{r_{\boldsymbol{x}}}{\sinh r_{\boldsymbol{x}}} \cdot \left( \cosh D_G(\boldsymbol{x}^H, \boldsymbol{y}^H) - 1 \right) \quad .
\end{aligned}
\tag{68}
$$

Note that eq. (68) is a negative-valued and concave distortion.

**Row V —** for $\mathfrak{X} = \mathbb{R}^d_{+*}$, consider $\varphi(\boldsymbol{x}) = \sum_i x_i \log x_i - x_i$ and $g(\boldsymbol{x}) = \mathbf{1}^\top \boldsymbol{x}$ (we normalize on the simplex). Since $g$ is linear, we do not need to check for the homogeneity of $\varphi$, and we directly obtain:

$$
\begin{aligned}
g(\boldsymbol{x}) \cdot D_\varphi \left( \frac{1}{g(\boldsymbol{x})} \cdot \boldsymbol{x} \middle\| \frac{1}{g(\boldsymbol{y})} \cdot \boldsymbol{y} \right) &= \sum_i x_i \log x_i - (\mathbf{1}^\top \boldsymbol{x}) \log(\mathbf{1}^\top \boldsymbol{x}) - \mathbf{1}^\top \boldsymbol{x} \\
&\quad - \frac{\mathbf{1}^\top \boldsymbol{x}}{\mathbf{1}^\top \boldsymbol{y}} \cdot \sum_i y_i \log y_i - (\mathbf{1}^\top \boldsymbol{x}) \log(\mathbf{1}^\top \boldsymbol{y}) + \mathbf{1}^\top \boldsymbol{x} \\
&\quad - (\mathbf{1}^\top \boldsymbol{x}) \cdot \sum_i \left( \frac{x_i}{\mathbf{1}^\top \boldsymbol{x}} - \frac{y_i}{\mathbf{1}^\top \boldsymbol{y}} \right) \cdot \log \frac{y_i}{\mathbf{1}^\top \boldsymbol{y}} \\
&= \sum_i x_i \log \frac{x_i}{y_i} - (\mathbf{1}^\top \boldsymbol{x}) \cdot \log \frac{\mathbf{1}^\top \boldsymbol{x}}{\mathbf{1}^\top \boldsymbol{y}} \quad .
\end{aligned}
\tag{69}
$$

Furthermore,

$$
\check{\varphi}(\boldsymbol{x}) = \mathbf{1}^\top \boldsymbol{x} \cdot \left( \sum_i \frac{x_i}{\mathbf{1}^\top \boldsymbol{x}} \cdot \log \frac{x_i}{\mathbf{1}^\top \boldsymbol{x}} - 1 \right) = \sum_i x_i \log x_i - (\mathbf{1}^\top \boldsymbol{x}) \log(\mathbf{1}^\top \boldsymbol{x}) - \mathbf{1}^\top \boldsymbol{x} \quad . \tag{70}
$$

Noting that $\check{\varphi}(\boldsymbol{x})$ is the sum of three terms, one of which is linear and can be removed for the divergence, so the divergence is just the sum of the two divergences with the two generators, which is found to be Equation (69) as well. Remark that while the KL divergence is convex in its both arguments, $D_{\check{\varphi}}(\boldsymbol{x} \| \boldsymbol{y})$ may not be (jointly) convex. Indeed, its Hessian in $\boldsymbol{y}$ equals:

$$
\mathrm{H}_{\boldsymbol{y}}(D_{\check{\varphi}}) = \mathrm{Diag}(\{x_i/y_i^2\}_i) - \frac{\mathbf{1}^\top \boldsymbol{x}}{(\mathbf{1}^\top \boldsymbol{y})^2} \cdot \mathbf{1}\mathbf{1}^\top \quad , \tag{71}
$$

which may be indefinite.

**Row VI —** for $\mathcal{X} = \mathbb{R}^d_{+*}$, consider $\varphi(\boldsymbol{x}) = -d - \sum_i \log x_i$ and $g(\boldsymbol{x}) = (\pi_{\boldsymbol{x}})^{1/d}$, where we let $\pi_{\boldsymbol{x}} \doteq \prod_i x_i$ (we normalize with the geometric average). It comes

$$\check{\varphi}(\boldsymbol{x}) = (\pi_{\boldsymbol{x}})^{1/d} \cdot \left( -d - \sum_i \log \frac{x_i}{(\pi_{\boldsymbol{x}})^{1/d}} \right) = -d \cdot (\pi_{\boldsymbol{x}})^{1/d} . \tag{72}$$

$g$ is not linear (but it is homogeneous of degree 1), and we have

$$\varphi(\boldsymbol{x}) = -d = \boldsymbol{x}^\top \nabla \varphi(\boldsymbol{x}) \ , \forall \boldsymbol{x} : \prod_i x_i = 1 \ , \tag{73}$$

so $\varphi$ is 1-homogeneous on $\mathcal{X}_g$, and we can apply Theorem 1. We have

$$g(\boldsymbol{x}) \cdot D_\varphi \left( \frac{1}{g(\boldsymbol{x})} \cdot \boldsymbol{x} \Big\| \frac{1}{g(\boldsymbol{y})} \cdot \boldsymbol{y} \right)$$

$$= (\pi_{\boldsymbol{x}})^{1/d} \cdot \sum_i \left( \frac{x_i(\pi_{\boldsymbol{y}})^{1/d}}{y_i(\pi_{\boldsymbol{x}})^{1/d}} - \log \frac{x_i(\pi_{\boldsymbol{y}})^{1/d}}{y_i(\pi_{\boldsymbol{x}})^{1/d}} \right) - d(\pi_{\boldsymbol{x}})^{1/d}$$

$$= \sum_i \frac{x_i(\pi_{\boldsymbol{y}})^{1/d}}{y_i} - d(\pi_{\boldsymbol{x}})^{1/d} \log(\pi_{\boldsymbol{x}})^{1/d} - (\pi_{\boldsymbol{x}})^{1/d} \log \pi_{\boldsymbol{y}} + (\pi_{\boldsymbol{x}})^{1/d} \log \pi_{\boldsymbol{y}}$$

$$+ d(\pi_{\boldsymbol{x}})^{1/d} \log(\pi_{\boldsymbol{x}})^{1/d} - d(\pi_{\boldsymbol{x}})^{1/d}$$

$$= \sum_i \frac{x_i(\pi_{\boldsymbol{y}})^{1/d}}{y_i} - d(\pi_{\boldsymbol{x}})^{1/d} \ . \tag{74}$$

We also have

$$\frac{\partial}{\partial x_i} \check{\varphi}(\boldsymbol{x}) = -(1/x_i) \cdot (\pi_{\boldsymbol{x}})^{1/d} \ , \tag{75}$$

and so

$$D_{\check{\varphi}}(\boldsymbol{x} \| \boldsymbol{y}) = -d(\pi_{\boldsymbol{x}})^{1/d} + d(\pi_{\boldsymbol{y}})^{1/d} + \cdot \sum_i (x_i - y_i) \cdot \frac{(\pi_{\boldsymbol{y}})^{1/d}}{y_i}$$

$$= -d(\pi_{\boldsymbol{x}})^{1/d} + d(\pi_{\boldsymbol{y}})^{1/d} + \sum_i \frac{x_i(\pi_{\boldsymbol{y}})^{1/d}}{y_i} - d(\pi_{\boldsymbol{y}})^{1/d}$$

$$= \sum_i \frac{x_i(\pi_{\boldsymbol{y}})^{1/d}}{y_i} - d(\pi_{\boldsymbol{x}})^{1/d} \ , \tag{76}$$

which is equal to Equation (74), so we check that Theorem 1 applies in this case.

**Row VII —** We use the following fact Kulis et al. [2009]. Let $\mathrm{X} = \mathrm{U L U}^\top$ and $\mathrm{Y} = \mathrm{V T V}^\top$ be the eigendecomposition of symmetric positive definite matrices $\mathrm{X}$ and $\mathrm{Y}$, with $\mathrm{L} \doteq \mathrm{Diag}(\boldsymbol{l})$, $\mathrm{T} \doteq \mathrm{Diag}(\boldsymbol{t})$, and $\mathrm{U} \doteq [\boldsymbol{u}_1 | \boldsymbol{u}_2 | \cdots | \boldsymbol{u}_d]$, $\mathrm{V} \doteq [\boldsymbol{v}_1 | \boldsymbol{v}_2 | \cdots | \boldsymbol{v}_d]$ orthonormal; let $\varphi = \mathrm{tr}\,(\mathrm{X} \log \mathrm{X} - \mathrm{X})$. Then we have

$$D_\varphi(\mathrm{X} \| \mathrm{Y}) = \sum_{i,j} (\boldsymbol{u}_i^\top \boldsymbol{v}_j)^2 \cdot D_{\varphi_2}(l_i \| t_j) \ , \tag{77}$$

with $\varphi_2(x) = x \log x - x$. We pick $g(\mathrm{X}) = \mathrm{tr}\,(\mathrm{X}) = \sum_i l_i$, which brings from Equation (69)

$$
g(\mathrm{X}) \cdot D_\varphi\left(\frac{1}{g(\mathrm{X})} \cdot \mathrm{X} \Big\| \frac{1}{g(\mathrm{Y})} \cdot \mathrm{Y}\right)
$$

$$
= \sum_{i,j} (\boldsymbol{u}_i^\top \boldsymbol{v}_j)^2 \cdot \mathrm{tr}\,(\mathrm{X}) \cdot D_{\varphi_2}\left(\frac{l_i}{\mathrm{tr}\,(\mathrm{X})} \Big\| \frac{t_j}{\mathrm{tr}\,(\mathrm{Y})}\right)
$$

$$
= \sum_{i,j} (\boldsymbol{u}_i^\top \boldsymbol{v}_j)^2 \cdot \mathrm{tr}\,(\mathrm{X}) \cdot \left(\frac{l_i}{\mathrm{tr}\,(\mathrm{X})} \cdot \log \frac{l_i \cdot \mathrm{tr}\,(\mathrm{Y})}{t_j \cdot \mathrm{tr}\,(\mathrm{X})} - \frac{l_i}{\mathrm{tr}\,(\mathrm{X})} + \frac{t_j}{\mathrm{tr}\,(\mathrm{Y})}\right)
$$

$$
= \sum_{i,j} (\boldsymbol{u}_i^\top \boldsymbol{v}_j)^2 \cdot \left(l_i \log \frac{l_i}{t_j} - l_i + t_j\right) + \log\left(\frac{\mathrm{tr}\,(\mathrm{Y})}{\mathrm{tr}\,(\mathrm{X})}\right) \cdot \sum_{i,j} (\boldsymbol{u}_i^\top \boldsymbol{v}_j)^2 \cdot l_i
$$

$$
+ \frac{\mathrm{tr}\,(\mathrm{X})}{\mathrm{tr}\,(\mathrm{Y})} \cdot \sum_{i,j} (\boldsymbol{u}_i^\top \boldsymbol{v}_j)^2 \cdot t_j - \sum_{i,j} (\boldsymbol{u}_i^\top \boldsymbol{v}_j)^2 \cdot t_j \quad . \tag{78}
$$

Because $\mathrm{U}$, $\mathrm{V}$ are orthonormal, we also get $\sum_{i,j} (\boldsymbol{u}_i^\top \boldsymbol{v}_j)^2 \cdot l_i = \sum_i l_i \sum_j \cos^2(\boldsymbol{u}_i, \boldsymbol{v}_j) = \sum_i l_i = \mathrm{tr}\,(\mathrm{X})$ and $\sum_{i,j} (\boldsymbol{u}_i^\top \boldsymbol{v}_j)^2 \cdot t_j = \mathrm{tr}\,(y)$, and so Equation (78) becomes

$$
g(\mathrm{X}) \cdot D_\varphi\left(\frac{1}{g(\mathrm{X})} \cdot \mathrm{X} \Big\| \frac{1}{g(\mathrm{Y})} \cdot \mathrm{Y}\right)
$$

$$
= \mathrm{tr}\,(\mathrm{X} \log \mathrm{X} - \mathrm{X} \log \mathrm{Y}) - \mathrm{tr}\,(\mathrm{X}) + \mathrm{tr}\,(\mathrm{Y}) + \mathrm{tr}\,(\mathrm{X}) \cdot \log\left(\frac{\mathrm{tr}\,(\mathrm{Y})}{\mathrm{tr}\,(\mathrm{X})}\right) + \mathrm{tr}\,(\mathrm{X}) - \mathrm{tr}\,(\mathrm{Y})
$$

$$
= \mathrm{tr}\,(\mathrm{X} \log \mathrm{X} - \mathrm{X} \log \mathrm{Y}) - \mathrm{tr}\,(\mathrm{X}) \cdot \log\left(\frac{\mathrm{tr}\,(\mathrm{X})}{\mathrm{tr}\,(\mathrm{Y})}\right) \quad . \tag{79}
$$

We also check that

$$
\check{\varphi}(\mathrm{X}) = \mathrm{tr}\,(\mathrm{X}) \cdot \mathrm{tr}\left(\frac{1}{\mathrm{tr}\,(\mathrm{X})} \cdot \mathrm{X} \log\left(\frac{1}{\mathrm{tr}\,(\mathrm{X})} \cdot \mathrm{X}\right) - \frac{1}{\mathrm{tr}\,(\mathrm{X})} \cdot \mathrm{X}\right)
$$

$$
= \mathrm{tr}\left(\mathrm{X} \log\left(\frac{1}{\mathrm{tr}\,(\mathrm{X})} \cdot \mathrm{X}\right)\right) - \mathrm{tr}\,(\mathrm{X}) \quad , \tag{80}
$$

and

$$
\mathrm{X} \log\left(\frac{1}{\mathrm{tr}\,(\mathrm{X})} \cdot \mathrm{X}\right) = \mathrm{U}\mathrm{L}\mathrm{U}^\top \mathrm{U} \log\left(\frac{1}{\mathbf{1}^\top \boldsymbol{l}} \cdot \mathrm{L}\right) \mathrm{U}^\top
$$

$$
= \mathrm{U}\mathrm{L} \log\left(\frac{1}{\mathbf{1}^\top \boldsymbol{l}} \cdot \mathrm{L}\right) \mathrm{U}^\top \tag{81}
$$

$$
= \mathrm{U}\mathrm{L} \log \mathrm{L}\mathrm{U}^\top - \log \mathrm{tr}\,(\mathrm{X}) \cdot \mathrm{U}\mathrm{L}\mathrm{U}^\top \quad , \tag{82}
$$

so that $\check{\varphi}(\mathrm{X}) = \mathrm{tr}\,(\mathrm{X} \log \mathrm{X} - \mathrm{X}) - \mathrm{tr}\,(\mathrm{X}) \cdot \log \mathrm{tr}\,(\mathrm{X})$. Let $\varphi_3(\mathrm{X}) \doteq \mathrm{tr}\,(\mathrm{X}) \cdot \log \mathrm{tr}\,(\mathrm{X})$. We have $\nabla \varphi_3(\mathrm{X}) = (1 + \log \mathrm{tr}\,(\mathrm{X})) \cdot \mathrm{I}$. Since a (Bregman) divergence involving a sum of generators is the sum of (Bregman)

divergences, we get

$$
\begin{aligned}
D_{\check{\varphi}}\left(\mathrm{X}\|\mathrm{Y}\right) &= \operatorname{tr}\left(\mathrm{X}\log\mathrm{X}-\mathrm{X}\log\mathrm{Y}-\mathrm{X}+\mathrm{Y}\right)-\operatorname{tr}\left(\mathrm{X}\right)\cdot\log\operatorname{tr}\left(\mathrm{X}\right)+\operatorname{tr}\left(\mathrm{Y}\right)\cdot\log\operatorname{tr}\left(\mathrm{Y}\right)\\
&\quad +(1+\log\operatorname{tr}\left(\mathrm{Y}\right))\cdot\operatorname{tr}\left(\mathrm{X}-\mathrm{Y}\right)\\
&= \operatorname{tr}\left(\mathrm{X}\log\mathrm{X}-\mathrm{X}\log\mathrm{Y}\right)-\operatorname{tr}\left(\mathrm{X}\right)\cdot\log\operatorname{tr}\left(\mathrm{X}\right)+\operatorname{tr}\left(\mathrm{X}\right)\cdot\log\operatorname{tr}\left(\mathrm{Y}\right)\\
&= \operatorname{tr}\left(\mathrm{X}\log\mathrm{X}-\mathrm{X}\log\mathrm{Y}\right)-\operatorname{tr}\left(\mathrm{X}\right)\cdot\log\left(\frac{\operatorname{tr}\left(\mathrm{X}\right)}{\operatorname{tr}\left(\mathrm{Y}\right)}\right)\ ,
\end{aligned}
\tag{83}
$$

which is Equation (79).

**Row VIII —** We have the same property as for Row V, but this time with $\varphi_2 = -d - \log x$ [Kulis et al., 2009]. We check that whenever $\det(\mathrm{X}) = 1$, we have

$$
\begin{aligned}
\varphi(\mathrm{X}) = -d - \log\det(\mathrm{X}) &= -d\\
&= -\det(\mathrm{X})\operatorname{tr}\left(\mathrm{I}\right)\\
&= \operatorname{tr}\left(\det(\mathrm{X})\mathrm{X}^{-1}\mathrm{X}\right) = \operatorname{tr}\left(\nabla\varphi(\mathrm{X})^{\top}\mathrm{X}\right)\ .
\end{aligned}
\tag{84}
$$

For $g(\mathrm{X}) \doteq \det\mathrm{X}^{1/d}$, we get:

$$
\begin{aligned}
\check{\varphi}(\mathrm{X}) &= \det\mathrm{X}^{1/d}\cdot\left(-d-\log\det\left(\frac{1}{\det\mathrm{X}^{1/d}}\cdot\mathrm{X}\right)\right)\\
&= \det\mathrm{X}^{1/d}\cdot\left(-d-\log\frac{1}{\det\mathrm{X}}\cdot\det\mathrm{X}\right) = -d\cdot\det\mathrm{X}^{1/d}\ ,
\end{aligned}
\tag{85}
$$

and furthermore

$$
\begin{aligned}
\nabla\check{\varphi}(\mathrm{X}) &= -d\cdot\nabla(\det\mathrm{X}^{1/d})(\mathrm{X})\\
&= -\det(\mathrm{X}^{1/d})\cdot\mathrm{X}^{-1}
\end{aligned}
\tag{86}
$$

So,

$$
\begin{aligned}
D_{\check{\varphi}}\left(\mathrm{X}\|\mathrm{Y}\right) &= -d\cdot\det\mathrm{X}^{1/d}+d\cdot\det\mathrm{Y}^{1/d}+\operatorname{tr}\left(\det(\mathrm{Y}^{1/d})\cdot\mathrm{Y}^{-1}(\mathrm{X}-\mathrm{Y})\right)\\
&= -d\cdot\det\mathrm{X}^{1/d}+d\cdot\det\mathrm{Y}^{1/d}+\det(\mathrm{Y}^{1/d})\operatorname{tr}\left(\mathrm{X}\mathrm{Y}^{-1}\right)-d\cdot\det\mathrm{Y}^{1/d}\\
&= \det(\mathrm{Y}^{1/d})\operatorname{tr}\left(\mathrm{X}\mathrm{Y}^{-1}\right)-d\cdot\det\mathrm{X}^{1/d}\ .
\end{aligned}
\tag{87}
$$

We check that it is equal to:

$$
\begin{aligned}
g(\mathrm{X})\cdot D_{\varphi}&\left(\frac{1}{g(\mathrm{X})}\cdot\mathrm{X}\|\frac{1}{g(\mathrm{Y})}\cdot\mathrm{Y}\right)\\
&= \det\mathrm{X}^{1/d}\cdot\sum_{i,j}(\boldsymbol{u}_i^{\top}\boldsymbol{v}_j)^2\cdot\left(\frac{l_i\det\mathrm{Y}^{1/d}}{t_j\det\mathrm{X}^{1/d}}-\log\frac{l_i\det\mathrm{Y}^{1/d}}{t_j\det\mathrm{X}^{1/d}}-d\right)\ .
\end{aligned}
\tag{88}
$$

To check it, we use the fact that, since $\mathrm{U}$ and $\mathrm{V}$ are orthonormal,

$$
\sum_{i,j} (\boldsymbol{u}_i^\top \boldsymbol{v}_j)^2 \cdot \log \frac{l_i \det \mathrm{Y}^{1/d}}{t_j \det \mathrm{X}^{1/d}}
$$

$$
= \sum_{i,j} (\boldsymbol{u}_i^\top \boldsymbol{v}_j)^2 \cdot \log l_i - \sum_{i,j} (\boldsymbol{u}_i^\top \boldsymbol{v}_j)^2 \cdot \log \det \mathrm{X}^{1/d}
$$

$$
+ \sum_{i,j} (\boldsymbol{u}_i^\top \boldsymbol{v}_j)^2 \cdot \log \det \mathrm{Y}^{1/d} - \sum_{i,j} (\boldsymbol{u}_i^\top \boldsymbol{v}_j)^2 \cdot \log t_j
$$

$$
= \underbrace{\sum_i \log l_i - d \cdot \log \det \mathrm{X}^{1/d}}_{=0} + \underbrace{d \cdot \log \det \mathrm{Y}^{1/d} - \sum_j \log l_j}_{=0} = 0 \ , \tag{89}
$$

which yields

$$
g(\mathrm{X}) \cdot D_\varphi \left( \frac{1}{g(\mathrm{X})} \cdot \mathrm{X} \Big\| \frac{1}{g(\mathrm{Y})} \cdot \mathrm{Y} \right) = \det \mathrm{X}^{1/d} \cdot \sum_{i,j} (\boldsymbol{u}_i^\top \boldsymbol{v}_j)^2 \cdot \left( \frac{l_i \det \mathrm{Y}^{1/d}}{t_j \det \mathrm{X}^{1/d}} - d \right)
$$

$$
= \det \mathrm{Y}^{1/d} \cdot \sum_{i,j} (\boldsymbol{u}_i^\top \boldsymbol{v}_j)^2 \cdot \frac{l_i}{t_j} - d \cdot \det \mathrm{X}^{1/d}
$$

$$
= \det \mathrm{Y}^{1/d} \cdot \mathrm{tr} \left( \mathrm{X} \mathrm{Y}^{-1} \right) - d \cdot \det \mathrm{X}^{1/d} \ , \tag{90}
$$

which is equal to Equation (87).

# IV Going deep: higher-order identities

We can generalize Theorem 1 to higher order identities. For this, consider $k > 0$ an integer, and let $g_1, g_2, ..., g_k : \mathcal{X} \to \mathbb{R}_*$ be a sequence of differentiable functions. For any $\ell, \ell' \in [k]_*$ such that $\ell \le \ell'$, we let $\tilde{g}_{\ell,\ell'}$ be defined recursively as:

$$\tilde{g}_{\ell,\ell'}(\boldsymbol{x}) \;\doteq\; \begin{cases} \tilde{g}_{\ell-1,\ell'}(\boldsymbol{x}) \cdot g_{\ell'-(\ell-1)}\left(\frac{1}{\tilde{g}_{\ell-1,\ell'}(\boldsymbol{x})} \cdot \boldsymbol{x}\right) & \text{if} \quad 1 < \ell \le \ell' \;, \\ g_{\ell'}(\boldsymbol{x}) & \text{if} \quad \ell = 1 \;, \end{cases} \tag{91}$$

and, for any $\ell \in [k]$,

$$\check{\varphi}^{(\ell)}(\boldsymbol{x}) \;\doteq\; \begin{cases} g_\ell(\boldsymbol{x}) \cdot \check{\varphi}^{(\ell-1)}\left(\frac{1}{g_\ell(\boldsymbol{x})} \cdot \boldsymbol{x}\right) & \text{if} \quad 0 < \ell \le k \;, \\ \varphi(\boldsymbol{x}) & \text{if} \quad \ell = 0 \;. \end{cases} \tag{92}$$

Notice that even when all $g.$ are affine, this does not guarantee that some $\tilde{g}_{\ell,\ell'}$ for $\ell \ne 1$ is going to be affine. However, if for example $g_{\ell'}$ is affine and all "preceeding" $g_\ell$ ($\ell \le \ell'$) are homogeneous of degree 1, then all $\tilde{g}_{\ell,\ell'}$ ($\forall \ell \le \ell'$) are affine. The following result can be seen as extension of the functional composition rules known for generalized perspective transforms of functions [Maréchal, 2005b] to composition rules for perspective transforms of divergences (See Section VIII).

**Corollary L** *For any $k \in \mathbb{N}_*$, let $\varphi : \mathcal{X} \to \mathbb{R}$ be convex differentiable, and $g_\ell : \mathcal{X} \to \mathbb{R}_*$ ($\ell \in [k]$) a sequence of $k$ differentiable functions. Then the following relationship holds, for any $\ell, \ell' \in [k]_*$ with $\ell \le \ell'$:*

$$\tilde{g}_{\ell,\ell'}(\boldsymbol{x}) \cdot D_{\check{\varphi}^{(\ell'-\ell)}}\left(\frac{1}{\tilde{g}_{\ell,\ell'}(\boldsymbol{x})} \cdot \boldsymbol{x} \Big\| \frac{1}{\tilde{g}_{\ell,\ell'}(\boldsymbol{y})} \cdot \boldsymbol{y}\right) \;=\; D_{\check{\varphi}^{(\ell')}}(\boldsymbol{x}\|\boldsymbol{y}) \;\;, \forall \boldsymbol{x}, \boldsymbol{y} \in \mathcal{X} \;, \tag{93}$$

*with $\tilde{g}_{\ell,\ell'}$ defined as in Equation (91) and $\check{\varphi}^{(\ell')}$ defined as in Equation (92), if and only if at least one of the two following conditions hold:*

   *(i) $\tilde{g}_{\ell,\ell'}$ is affine on $\mathcal{X}$;*

   *(ii) $\check{\varphi}^{(\ell'-\ell)}$ is positive homogeneous of degree 1 on $\mathcal{X}_{\ell,\ell'} \doteq \{(1/\tilde{g}_{\ell,\ell'}(\boldsymbol{x})) \cdot \boldsymbol{x} : \boldsymbol{x} \in \mathcal{X}\}$.*

We check that whenever $\varphi$ is convex and all $g.$ are non-negative, then all $\check{\varphi}^{(\ell)}$ are convex ($\forall \ell \in [k]$). To prove this, we choose $\ell' = \ell$ and rewrite Equation (3), which brings, since $\check{\varphi}^{(\ell'-\ell)} = \check{\varphi}^{(0)} = \varphi$,

$$\tilde{g}_{\ell,\ell}(\boldsymbol{x}) \cdot D_\varphi\left(\frac{1}{\tilde{g}_{\ell,\ell}(\boldsymbol{x})} \cdot \boldsymbol{x} \Big\| \frac{1}{\tilde{g}_{\ell,\ell}(\boldsymbol{y})} \cdot \boldsymbol{y}\right) \;=\; D_{\check{\varphi}^{(\ell)}}(\boldsymbol{x}\|\boldsymbol{y}) \;\;, \forall \boldsymbol{x}, \boldsymbol{y} \in \mathcal{X} \;. \tag{94}$$

Since $\varphi$ is convex, a sufficient condition to prove our result is to show that $\tilde{g}_{\ell,\ell}$ is non-negative — which will prove that the right hand side of (94) is non-negative, and therefore $\check{\varphi}^{(\ell)}$ is convex —. This can easily be proven by induction from the expression of $\tilde{g}_{\ell,\ell'}$ in (91) and the fact that all $g.$ are non-negative.

One interesting candidate for simplification is when all $g_{\cdot}$ are the same affine function, say $g_\ell(\boldsymbol{x}) = \boldsymbol{a}^\top \boldsymbol{x} + b, \forall \ell \in [k]$. In this case, we have indeed:

$$
\begin{aligned}
\tilde{g}_{\ell,\ell'}(\boldsymbol{x}) &= \boldsymbol{a}^\top \boldsymbol{x} + b \cdot \tilde{g}_{\ell-1,\ell'}(\boldsymbol{x}) \\
&= b^\ell + \boldsymbol{a}^\top \boldsymbol{x} \cdot \sum_{j=1}^{\ell-1} b^j \ , \tag{95}
\end{aligned}
$$

$$
\check{\varphi}^{(\ell')}(\boldsymbol{x}) = \left( b^{\ell'} + \boldsymbol{a}^\top \boldsymbol{x} \cdot \sum_{j=1}^{\ell'-1} b^j \right) \cdot \varphi \left( \frac{1}{b^{\ell'} + \boldsymbol{a}^\top \boldsymbol{x} \cdot \sum_{j=1}^{\ell'-1} b^j} \cdot \boldsymbol{x} \right) \ . \tag{96}
$$

**Proof** [Proof of Corollary L] To check eq. (4), we first remark ($\ell'$ being fixed) that it holds for $\ell = 1$ (this is eq. (4)), and then proceed by an induction from the induction base hypothesis that, for some $\ell \leq \ell'$,

$$
\check{\varphi}^{(\ell')}(\boldsymbol{x}) = \tilde{g}_{\ell,\ell'}(\boldsymbol{x}) \cdot \check{\varphi}^{(\ell'-\ell)} \left( \frac{1}{\tilde{g}_{\ell,\ell'}(\boldsymbol{x})} \cdot \boldsymbol{x} \right) \ . \tag{97}
$$

We now have

$$
\begin{aligned}
&\check{\varphi}^{(\ell')}(\boldsymbol{x}) \\
&= \frac{\tilde{g}_{\ell+1,\ell'}(\boldsymbol{x})}{g_{\ell'-\ell} \left( \frac{1}{\tilde{g}_{\ell,\ell'}(\boldsymbol{x})} \cdot \boldsymbol{x} \right)} \cdot \check{\varphi}^{(\ell'-\ell)} \left( \frac{1}{\tilde{g}_{\ell,\ell'}(\boldsymbol{x})} \cdot \boldsymbol{x} \right) \tag{98} \\
&= \frac{\tilde{g}_{\ell+1,\ell'}(\boldsymbol{x})}{g_{\ell'-\ell} \left( \frac{1}{\tilde{g}_{\ell,\ell'}(\boldsymbol{x})} \cdot \boldsymbol{x} \right)} \cdot g_{\ell'-\ell} \left( \frac{1}{\tilde{g}_{\ell,\ell'}(\boldsymbol{x})} \cdot \boldsymbol{x} \right) \cdot \check{\varphi}^{(\ell'-(\ell+1))} \left( \frac{1}{\tilde{g}_{\ell,\ell'}(\boldsymbol{x}) g_{\ell'-\ell} \left( \frac{1}{\tilde{g}_{\ell,\ell'}(\boldsymbol{x})} \cdot \boldsymbol{x} \right)} \cdot \boldsymbol{x} \right) \tag{99} \\
&= \tilde{g}_{\ell+1,\ell'}(\boldsymbol{x}) \cdot \check{\varphi}^{(\ell'-(\ell+1))} \left( \frac{1}{\tilde{g}_{\ell,\ell'}(\boldsymbol{x}) g_{\ell'-\ell} \left( \frac{1}{\tilde{g}_{\ell,\ell'}(\boldsymbol{x})} \cdot \boldsymbol{x} \right)} \cdot \boldsymbol{x} \right) \\
&= \tilde{g}_{\ell+1,\ell'}(\boldsymbol{x}) \cdot \check{\varphi}^{(\ell'-(\ell+1))} \left( \frac{1}{\tilde{g}_{\ell+1,\ell'}(\boldsymbol{x})} \cdot \boldsymbol{x} \right) \ . \tag{100}
\end{aligned}
$$

Eq. (98) comes from eq. (97) and the definition of $\tilde{g}_\ell$ in (91), eq. (99) comes from the definition of $\check{\varphi}^{(\ell'-\ell)}$ in (92), eq. (100) is a second use of the definition of $\tilde{g}_\ell$ in (91). ∎

Notice the eventual high non-linearities introduced by the composition in eqs (91,92), which justifies the "deep" characterization.

# V  Additional application: perspective transform of exponential families

Let $\varphi$ be the cumulant function of a regular $\varphi$-exponential family with pdf $p_\varphi(.|\boldsymbol{\theta})$, where $\boldsymbol{\theta} \in \mathfrak{X}$ is its natural parameter. Let $\Omega(.)$ be a norm on $\mathfrak{X}$. Let $\boldsymbol{\theta}_\Omega$ be the image of $\boldsymbol{\theta} \in \mathfrak{X}$ by the application from $\mathfrak{X}$ onto the $\Omega$-ball of unit norm defined by $\boldsymbol{x} \mapsto (1/\Omega(\boldsymbol{x})) \cdot \boldsymbol{x}$. For any two $\boldsymbol{\theta}, \boldsymbol{\theta}' \in \mathfrak{X}$, let

$$\mathrm{KL}_\varphi(\boldsymbol{\theta}\|\boldsymbol{\theta}') \;\doteq\; \int p_\varphi(\boldsymbol{x}|\boldsymbol{\theta}) \log \frac{p_\varphi(\boldsymbol{x}|\boldsymbol{\theta})}{p_\varphi(\boldsymbol{x}|\boldsymbol{\theta}')} \mathrm{d}\boldsymbol{x} \tag{101}$$

be the KL divergence between the two densities $p_\varphi(.|\boldsymbol{\theta})$ and $p_\varphi(.|\boldsymbol{\theta}')$. A function is called regular if $\Phi \doteq \{\boldsymbol{\theta} : \varphi(\boldsymbol{\theta}) \ll \infty\}$ is open.

**Lemma M** *(perspective transform of exponential families) For any convex regular $\varphi$ which is restricted positive 1-homogeneous on $\mathfrak{X}_\Omega$, the KL-divergence between two members of the same $\varphi$-exponential family satisfies:*

$$\Omega(\boldsymbol{\theta}') \cdot \mathrm{KL}_\varphi(\boldsymbol{\theta}_\Omega\|\boldsymbol{\theta}'_\Omega) = D_{\check{\varphi}}(\boldsymbol{\theta}'\|\boldsymbol{\theta}) = \mathrm{KL}_{\check{\varphi}}(\boldsymbol{\theta}\|\boldsymbol{\theta}') \;. \tag{102}$$

**Proof**  We know that $\mathrm{KL}(\boldsymbol{\theta}\|\boldsymbol{\theta}') = D_\varphi(\boldsymbol{\theta}'\|\boldsymbol{\theta})$ [Boissonnat et al., 2010]. Hence,

$$\begin{aligned}
D_{\check{\varphi}}(\boldsymbol{\theta}'\|\boldsymbol{\theta}) &= \Omega(\boldsymbol{\theta}') \cdot D_\varphi \left( \frac{1}{\Omega(\boldsymbol{\theta}')} \cdot \boldsymbol{\theta}' \| \frac{1}{\Omega(\boldsymbol{\theta})} \cdot \boldsymbol{\theta} \right) \\
&= \Omega(\boldsymbol{\theta}') \cdot D_\varphi(\boldsymbol{\theta}'_\Omega\|\boldsymbol{\theta}_\Omega) \\
&= \Omega(\boldsymbol{\theta}') \cdot \mathrm{KL}_\varphi(\boldsymbol{\theta}_\Omega\|\boldsymbol{\theta}'_\Omega) \;,
\end{aligned} \tag{103}$$

as claimed. To prove the rest of the Lemma, we remark that $\check{\Phi}$ is open because $\Phi$ is open and $\check{\varphi}$ is convex because $\Omega(\boldsymbol{\theta}) \geq 0$ and $\varphi$ is convex, so $\check{\varphi}$ is convex regular and defines the cumulant of a regular exponential family for which $D_{\check{\varphi}}(\boldsymbol{\theta}'\|\boldsymbol{\theta}) = \mathrm{KL}_{\check{\varphi}}(\boldsymbol{\theta}\|\boldsymbol{\theta}')$ [Banerjee et al., 2005]. ∎

The interest in Lemma M is to provide an integral-free expression of the KL-divergence when natural parameters are scaled by non-trivial transformations (left inequality). Furthermore, the equality

$$\Omega(\boldsymbol{\theta}') \cdot \mathrm{KL}_\varphi \left( \frac{1}{\Omega(\boldsymbol{\theta})} \cdot \boldsymbol{\theta} \,\Big\|\, \frac{1}{\Omega(\boldsymbol{\theta}')} \cdot \boldsymbol{\theta}' \right) \;=\; \mathrm{KL}_{\check{\varphi}}(\boldsymbol{\theta}\|\boldsymbol{\theta}') \tag{104}$$

states a valid generalized perspective transform equality because $\Omega$ is proper convex [Maréchal, 2005a,b]. Notice however that the rescaling of the KL divergence on the left uses its *right* parameter, unlike in Theorem 1. To summarize the content of Lemma M, we first "polarize" (left / right) the perspective transform of a Bregman divergence as a reference of which parameter is used to rescale the divergence. We also define as the *perspective transform of an exponential family* as the new distribution whose cumulant is the perspective transform of the cumulant (we do not change the natural parameter). We can then summarize eq. (104) by:

> "*the right perspective transform of the KL divergence between two distributions of the same exponential family is the KL divergence between the perspective transform of the distributions*"

Finally, it is out of the scope of this paper, but Lemma M can also be extended to generalized exponential families [Fongillo and Reid, 2013].

# VI  Additional application: computational information geometry

Two important objects of central importance in (computational) geometry are balls and Voronoi diagrams induced by a distortion, with which we can characterize the topological and computational aspects of major structures (Voronoi diagrams, triangulations, nearest neighbor topologies, etc.) [Boissonnat et al., 2010].

## VI.1  Bregman balls

Since a Bregman divergence is not necessarily symmetric, there are two types of (dual) balls that can be defined, the first or second types, where the variable $\boldsymbol{x}$ is respectively placed in the left or right position. The first type Bregman balls are convex while the second type are not necessarily convex. A (closed) Bregman ball of the second type (with center $\boldsymbol{c}$ and "radius" $r$) is defined as:

$$B'(\boldsymbol{c}, r | \mathfrak{X}, \varphi) \;\doteq\; \{\boldsymbol{x} \in \mathfrak{X} : D_\varphi(\boldsymbol{c} \| \boldsymbol{x}) \leq r\} \ . \tag{105}$$

It turns out that any divergence $D_{\check{\varphi}}$ induces a ball of the second type, which is not necessarily analytically a Bregman ball (when $\check{\varphi}$ is not convex), *but* turns out to define the *same* ball as a Bregman ball over properly scaled arguments (notice that the scaling is the same for both arguments, the ball's center and radius).

**Theorem N**  *Let $(\varphi, g, \check{\varphi})$ satisfy the conditions of Theorem 1, with $g$ non negative and $g(\boldsymbol{c}) \neq 0$. Then*

$$B'(\boldsymbol{c}, r | \check{\varphi}, \mathfrak{X}) \;=\; B'\left(\frac{1}{g(\boldsymbol{c})} \cdot \boldsymbol{c}, \frac{r}{g(\boldsymbol{c})} \,\middle|\, \varphi, \mathfrak{X}_g\right) \ . \tag{106}$$

**Proof**  From Theorem 1, we have

$$D_{\check{\varphi}}(\boldsymbol{c} \| \boldsymbol{x}) \;\leq\; r \tag{107}$$

iff

$$D_\varphi\left(\frac{1}{g(\boldsymbol{c})} \cdot \boldsymbol{c} \,\middle\|\, \frac{1}{g(\boldsymbol{c})} \cdot \boldsymbol{x}\right) \;\leq\; \frac{1}{g(\boldsymbol{c})} \cdot r \ . \tag{108}$$

Hence,

$$
\begin{aligned}
B'(\boldsymbol{c}, r | \check{\varphi}, \mathfrak{X}) \;&=\; \{\boldsymbol{x} \in \mathfrak{X} : D_{\check{\varphi}}(\boldsymbol{c} \| \boldsymbol{x}) \leq r\} \\
&=\; \left\{\boldsymbol{x} \in \mathfrak{X} : D_\varphi\left(\frac{1}{g(\boldsymbol{c})} \cdot \boldsymbol{c} \,\middle\|\, \frac{1}{g(\boldsymbol{x})} \cdot \boldsymbol{x}\right) \leq \frac{r}{g(\boldsymbol{c})}\right\} \\
&=\; B'\left(\frac{1}{g(\boldsymbol{c})} \cdot \boldsymbol{c}, \frac{r}{g(\boldsymbol{c})} \,\middle|\, \varphi, \mathfrak{X}_g\right) \ ,
\end{aligned}
\tag{109}
$$

as claimed.  ∎

In other words and to be a little bit more specific,

*"any $\boldsymbol{x}$ belongs to the ball of the second type induced by $D_{\check{\varphi}}$ over $\mathfrak{X}$ **iff** $(1/g(\boldsymbol{x})) \cdot \boldsymbol{x}$ ($\in \mathfrak{X}_g$) belongs to the Bregman ball of the second type induced by $D_{\varphi}$ over $\mathfrak{X}_g$ (obtained by scaling both the center and radius by $g(\boldsymbol{c})$)."*

This property is not true for balls of the first type. What Theorem N says is that the topology induced by $D_{\check{\varphi}}$ over $\mathfrak{X}$ is just *no different* from that induced by $D_{\varphi}$ over $\mathfrak{X}_g$.

## VI.2 Bregman Voronoi diagrams

Let us now investigate Bregman Voronoi diagrams. In the same way as there exists two types of Bregman balls, we can define two types of Bregman Voronoi diagrams that depend on the equation of the *Bregman bisector* [Boissonnat et al., 2010]. Of particular interest is the Bregman bisector of the *first* type:

$$BB_{\varphi}(\boldsymbol{x}, \boldsymbol{y}|\mathfrak{X}) \;\; = \;\; \{\boldsymbol{z} \in \mathfrak{X} : D_{\varphi}(\boldsymbol{z}\|\boldsymbol{x}) = D_{\varphi}(\boldsymbol{z}\|\boldsymbol{y})\} \; . \tag{110}$$

Let us define $\boldsymbol{x}, \boldsymbol{y}$ as the Bregman bisector parameters. It turns out that any divergence $D_{\check{\varphi}}$ induces a bisector of the first type which is not necessarily analytically a Bregman bisector (when $\check{\varphi}$ is not convex), *but* turns out to define the *same* bisector as a Bregman bisector over transformed coordinates.

**Theorem O** *Let $(\varphi, g, \check{\varphi})$ satisfy the conditions of Theorem 1. Then*

$$BB_{\check{\varphi}}(\boldsymbol{x}, \boldsymbol{y}|\mathfrak{X}) \;\; = \;\; BB_{\varphi}(\boldsymbol{x}, \boldsymbol{y}|\mathfrak{X}_g) \; . \tag{111}$$

(proof similar to Theorem N) Again, we get more precisely

*"any $\boldsymbol{x}$ belongs to a Bregman bisector of the first type induced by $D_{\check{\varphi}}$ over $\mathfrak{X}$ **iff** $(1/g(\boldsymbol{x})) \cdot \boldsymbol{x}$ ($\in \mathfrak{X}_g$) belongs to the corresponding Bregman bisector of the first type induced by $D_{\varphi}$ over $\mathfrak{X}_g$ (obtained by scaling both bisector parameters by $g(.)$)."*

This property is not true for Bregman bisectors of the second type (obtained by permuting $\boldsymbol{z}$ with the Bregman bisector parameters in eq . (110)).

## VI.3 Consequences

Theorems N, O have several important algorithmic consequences, some of which are listed now:

- the Voronoi diagram (resp. Delaunay triangulation) of the first type associated to $\check{\varphi}$ can be constructed via the Voronoi diagram (resp. Delaunay triangulation) of the first type associated to $\varphi$ [Boissonnat et al., 2010];

- range search using ball trees on $D_{\check{\varphi}}$ can be efficiently implemented using Bregman divergence $D_{\varphi}$ on $\mathfrak{X}_g$ [Cayton, 2009];

- the minimum enclosing ball problem, the one-class clustering problem (an important problem in machine learning), with balls of the second type on $D_{\check{\varphi}}$ can be solved via the minimum Bregman enclosing ball problem on $D_{\varphi}$ [Nock and Nielsen, 2005].

# VII Review: binary density ratio estimation

For completeness, we quickly review the central result of Menon and Ong [2016, Proposition 3]. Let $(P, Q, \pi)$ be densities giving $\mathbb{P}(\mathsf{X}|\mathsf{Y} = 1), \mathbb{P}(\mathsf{X} = \boldsymbol{x}|\mathsf{Y} = -1), \mathbb{P}(\mathsf{Y} = 1)$ respectively, and $M$ giving $\mathbb{P}(\mathsf{X} = \boldsymbol{x})$ accordingly. Let $r(\boldsymbol{x}) \doteq \mathbb{P}(\mathsf{X} = \boldsymbol{x}|\mathsf{Y} = 1)/\mathbb{P}(\mathsf{X} = \boldsymbol{x}|\mathsf{Y} = -1)$ be the density ratio of the class-conditional densities, and $\eta(\boldsymbol{x}) \doteq \mathbb{P}[\mathsf{Y} = 1|\mathsf{X} = \boldsymbol{x}]$ be the class-probability function. Then, we have the following, which extends [Menon and Ong, 2016, Proposition 6] for the case $\pi \neq \frac{1}{2}$.

**Lemma P** *Given a class-probability estimator $\hat{\eta}\colon \mathcal{X} \to [0, 1]$, let the density ratio estimator $\hat{r}$ be*

$$\hat{r}(\boldsymbol{x}) \;=\; \frac{1 - \pi}{\pi} \cdot \frac{\hat{\eta}(\boldsymbol{x})}{1 - \hat{\eta}(\boldsymbol{x})} \;. \tag{112}$$

*Then for any convex differentiable $\varphi\colon [0, 1] \to \mathbb{R}$,*

$$\mathbb{E}_{\mathsf{X} \sim M}[D_\varphi(\eta(\mathsf{X}) \| \hat{\eta}(\mathsf{X}))] \;=\; \pi \cdot \mathbb{E}_{\mathsf{X} \sim Q}\left[D_{\check{\varphi}}(r(\mathsf{X}) \| \hat{r}(\mathsf{X}))\right] \;. \tag{113}$$

*where $\check{\varphi}$ is as per Equation 4 with $g(z) \doteq \frac{1-\pi}{\pi} + z$ .*

**Proof** [Proof of Lemma P] Note that

$$
\begin{aligned}
\frac{1}{g(r(\boldsymbol{x}))} \cdot r(\boldsymbol{x}) \;&=\; \frac{\pi \mathbb{P}(\mathsf{X} = \boldsymbol{x}|\mathsf{Y} = -1)}{\mathbb{P}(\mathsf{X} = \boldsymbol{x})} \cdot \frac{\mathbb{P}(\mathsf{X} = \boldsymbol{x}|\mathsf{Y} = 1)}{\mathbb{P}(\mathsf{X} = \boldsymbol{x}|\mathsf{Y} = -1)} \\
&=\; \frac{\pi \mathbb{P}(\mathsf{X} = \boldsymbol{x}|\mathsf{Y} = 1)}{\mathbb{P}(\mathsf{X} = \boldsymbol{x})} \\
&=\; \eta(\boldsymbol{x}) \;,
\end{aligned}
\tag{114}
$$

and furthermore

$$
\begin{aligned}
\mathbb{P}(\mathsf{X} = \boldsymbol{x}) \;&=\; (1 - \pi)\mathbb{P}(\mathsf{X} = \boldsymbol{x}|\mathsf{Y} = -1) + \pi \mathbb{P}(\mathsf{X} = \boldsymbol{x}|\mathsf{Y} = 1) \\
&=\; \pi \cdot \left(\frac{1 - \pi}{\pi} + \frac{\mathbb{P}(\mathsf{X} = \boldsymbol{x}|\mathsf{Y} = 1)}{\mathbb{P}(\mathsf{X} = \boldsymbol{x}|\mathsf{Y} = -1)}\right) \cdot \mathbb{P}(\mathsf{X} = \boldsymbol{x}|\mathsf{Y} = -1) \\
&=\; \pi \cdot g(r(\boldsymbol{x})) \cdot \mathbb{P}(\mathsf{X} = \boldsymbol{x}|\mathsf{Y} = -1) \;.
\end{aligned}
\tag{115}
$$

So,

$$
\begin{aligned}
\mathbb{E}_{\mathsf{X} \sim M}[D_\varphi(\eta(\mathsf{X}) \| \hat{\eta}(\mathsf{X}))] \;&=\; \pi \cdot \mathbb{E}_{\mathsf{X} \sim Q}\left[g(r(\mathsf{X})) \cdot D_\varphi(\eta(\mathsf{X}) \| \hat{\eta}(\mathsf{X}))\right] & (116) \\
&=\; \pi \cdot \mathbb{E}_{\mathsf{X} \sim Q}\left[g(r(\mathsf{X})) \cdot D_\varphi\left(\frac{1}{g(r(\mathsf{X}))} \cdot r(\mathsf{X}) \,\Big\|\, \hat{\eta}(\mathsf{X})\right)\right] & (117) \\
&=\; \pi \cdot \mathbb{E}_{\mathsf{X} \sim Q}\left[g(r(\mathsf{X})) \cdot D_\varphi\left(\frac{1}{g(r(\mathsf{X}))} \cdot r(\mathsf{X}) \,\Big\|\, \frac{1}{g(\hat{r}(\mathsf{X}))} \cdot \hat{r}(\mathsf{X})\right)\right] & (118) \\
&=\; \pi \cdot \mathbb{E}_{\mathsf{X} \sim Q}\left[D_{\check{\varphi}}(r(\mathsf{X}) \| \hat{r}(\mathsf{X}))\right] \;, & (119)
\end{aligned}
$$

as claimed. Equation (116) comes from (115), Equation (117) comes from (114), Equation (118) comes from (112) and the definition of $g$. Equation (119) comes from Theorem 1, noting that $g$ is linear. ∎

Figure 3: A depiction of the adaptive isometry that Theorem 1 provides. To simplify the picture as much as possible, we have used the shorthands $\check{\boldsymbol{x}} \doteq (1/g(\boldsymbol{x})) \cdot \boldsymbol{x}$, $\check{\mathcal{X}} \doteq \mathcal{X}_g$. Double bars mean same metric length with respect to the (square root of) the Hessians in parenthesis.

# VIII   Comments on Theorem 1

## VIII.1   Theorem 1 vs scaled isometries

Theorem 1 states in fact an isometry under some conditions, but an adaptive one in the sense that metrics involved rely on all parameters, and in particular on the points involved in the divergences (See Figure 3). Indeed, a simple Taylor expansion of the equation (2) (main file) shows that any such Bregman distortion with a twice differentiable generator can be expressed as:

$$D_\varphi(\boldsymbol{x}\|\boldsymbol{y}) \;=\; \frac{1}{2} \cdot (\boldsymbol{x} - \boldsymbol{y})^\top \mathbf{H}_\varphi (\boldsymbol{x} - \boldsymbol{y}) \;, \tag{120}$$

for *some* value of the Hessian $\mathbf{H}_\varphi$ depending on $\boldsymbol{x}, \boldsymbol{y}$ (see for example [Kivinen et al., 2006, Appendix I], [Amari and Nagaoka, 2000]). Hence, under the constraint that both $\varphi$ and $\check{\varphi}$ are twice differentiable, eq. (3) becomes

$$g(\boldsymbol{x}) \cdot \left( \frac{1}{g(\boldsymbol{x})} \cdot \boldsymbol{x} - \frac{1}{g(\boldsymbol{y})} \cdot \boldsymbol{y} \right)^\top \mathbf{H}_\varphi \left( \frac{1}{g(\boldsymbol{x})} \cdot \boldsymbol{x} - \frac{1}{g(\boldsymbol{y})} \cdot \boldsymbol{y} \right) \;=\; (\boldsymbol{x} - \boldsymbol{y})^\top \mathbf{H}_{\check{\varphi}} (\boldsymbol{x} - \boldsymbol{y}) \;. \tag{121}$$

Notice that eq. (121) holds even when $\mathbf{H}_{\check{\varphi}}$ is indefinite. Assuming $g$ non-negative (which, by the way, enforces the convexity of $\check{\varphi}$ and prevents $\mathbf{H}_{\check{\varphi}}$ from being indefinite), we get by taking square roots,

$$\sqrt{g(\boldsymbol{x})} \cdot \left\| \frac{1}{g(\boldsymbol{x})} \cdot \boldsymbol{x} - \frac{1}{g(\boldsymbol{y})} \cdot \boldsymbol{y} \right\|_{\mathbf{H}_{\varphi}} = \|\boldsymbol{x} - \boldsymbol{y}\|_{\mathbf{H}_{\check{\varphi}}} \quad , \tag{122}$$

which is a scaled isometry relationship between $\mathfrak{X}_g$ (left) and $\mathfrak{X}$ (right), but again the metrics involved depend on the arguments. Nevertheless, eq. (122) displays a sophisticated relationship between distances in $\mathfrak{X}_g$ and in $\mathfrak{X}$ which may prove useful in itself. With this in mind and keeping into account the restrictions on $g$, Theorem 1 states via eq. (122) that

*"distances on $\mathfrak{X}$ with metric $\boldsymbol{H}_{\check{\varphi}}^{1/2}$ equal scaled distances after mapping $\boldsymbol{x} \mapsto (1/g(\boldsymbol{x})) \cdot \boldsymbol{x} \ (\in \mathfrak{X}_g)$ with metric $\boldsymbol{H}_{\varphi}^{1/2}$"*.

## VIII.2  Theorem 1 vs generalized perspective transforms

Perspective transforms, also defined as epi-multiplication [Bauschke et al., 2008], are well known objects in convex analysis, and used in machine learning in particular to design and analyse loss functions [Reid and Williamson, 2011]. [Maréchal, 2005a,b] has defined a generalized notion of perspective transforms which coincidentally happens to define $\check{\varphi}$ when assumptions are made about $g$. More precisely, Maréchal's generalized perspective transform of functions $\varphi$ and $g$ is defined as:

$$(\varphi \bigtriangleup g)(\boldsymbol{x}, \boldsymbol{y}) \ \doteq \ \begin{cases} g(\boldsymbol{y}) \cdot \varphi\left(\frac{1}{g(\boldsymbol{y})} \cdot \boldsymbol{x}\right) & \text{if} \quad g(\boldsymbol{y}) \in (0, +\infty) \\ \varphi 0^+(\boldsymbol{x}) & \text{if} \quad g(\boldsymbol{y}) = 0 \\ \infty & \text{if} \quad g(\boldsymbol{y}) = +\infty \end{cases} \tag{123}$$

where $\varphi 0^+$ is the recession function of $\varphi$. For the definition to be valid, both $\varphi$ and $g$ have to be proper convex and $g$ has to be positive. In this case, one remarks that

$$\check{\varphi}(\boldsymbol{x}) \ = \ (\varphi \bigtriangleup g)(\boldsymbol{x}, \boldsymbol{x}) \quad , \tag{124}$$

but of course this holds only when significant restrictions are put on $g$. This does not prevent very interesting cases for the application of Theorem 1, as witnessed by the application to exponential families given in Section V, as well as several examples in Table A1 (rows I, II, III, V, VII). In this case, Theorem 1 gives an indication of how to define the generalized perspective transform of a Bregman divergence, which would be just the left hand side of eq. (4) in Theorem 1. To our knowledge, the use perspective transforms in machine learning has been limited to the definition of Csiszar's duals for loss function and divergences [Reid and Williamson, 2011], and they have not been used to define the perspective of a divergence. Our results indicate that such objects would not be just mathematical curiosities, but could eventually be the ground for new methods to deal with popular problems. This is out of the scope of this paper, but if we resort to perspectives, then Theorem 1 can be roughly summarized by the property that

*"the perspective transform of the divergence equals the divergence of the perspective transform"*.

Figure 4: Density ratio estimate divergence $\mathbb{E}_{\mathsf{X} \sim P_C} \left[ D_{\check{\varphi}}(r(\mathsf{X}), \hat{r}(\mathsf{X})) \right]$ as a function of # of training samples.

# IX  Additional experiments: Multiclass density ratio experiments

We consider a synthetic multiclass density ratio estimation problem. We fix $\mathcal{X} = \mathbb{R}^2$, and consider $C = 3$ classes. We consider a distribution where the class-conditionals $\Pr(\mathsf{X}|\mathsf{Y} = c)$ are multivariate Gaussians with means $\boldsymbol{\mu}_c$ and covariance $\sigma_c^2 \cdot \mathrm{Id}$. As the class-conditionals have a closed form, we can explicitly compute $\boldsymbol{\eta}$, as well the density ratio $r$ to the reference class $c^* = C$.

For fixed class prior $\boldsymbol{\pi} = \Pr(\mathsf{Y} = c)$, we draw $N_{\mathrm{Tr}}$ samples from $\Pr(\mathsf{X}, \mathsf{Y})$. From this, we estimate the class-probability $\hat{\boldsymbol{\eta}}$ using multiclass logistic regression. This can be seen as minimising $\mathbb{E}_{\mathsf{X} \sim M} \left[ D_{\varphi}(\boldsymbol{\eta}(\mathsf{X}) \| \hat{\boldsymbol{\eta}}(\mathsf{X})) \right]$ where $\varphi(z) = \sum_i z_i \log z_i$ is the generator for the KL-divergence.

We then use Equation 6 (main file) to estimate the density ratios $\hat{r}$ from $\hat{\boldsymbol{\eta}}$. On a fresh sample of $N_{\mathrm{te}}$ instances from $\Pr(\mathsf{X}, \mathsf{Y})$, we estimate the right hand side of Lemma 2, *viz.* $\mathbb{E}_{\mathsf{X} \sim P_C} \left[ D_{\check{\varphi}}(r(\mathsf{X}) \| \hat{r}(\mathsf{X})) \right]$, where $\check{\varphi}$ uses the $g$ as specified in Lemma 2. From the result of Lemma 2, we expect this divergence to be small when $\hat{\boldsymbol{\eta}}$ is a good estimator of $\boldsymbol{\eta}$.

We perform the above for sample sizes $N \in \{4^4, 4^5, \ldots, 4^{10}\}$, with $N_{\mathrm{Tr}} = 0.8N$ and $N_{\mathrm{Te}} = 0.2N$. For each sample size, we perform $T = 25$ trials, where in each trial we randomly draw $\boldsymbol{\pi}$ uniformly over $(1/C)\mathbf{1} + (1 - 1/C) \cdot [0, 1]^C$, $\boldsymbol{\mu}_c$ from $0.1 \cdot \mathcal{N}(\mathbf{0}, 1)$, and $\sigma_c$ uniformly from $[0.5, 1]$. Figure 4 summarises the mean divergence across the $T$ trials for each sample size. We see that, as expected, with more training samples the divergence decreases in a monotone fashion.

# X  Additional experiments: Adaptive filtering experiments

Tables A2 – A7 present *in extenso* the experiments of $p$-LMS vs DN-$p$-LMS, as a function of $(p, q)$, whether target $\boldsymbol{u}$ is sparse or not, and the misestimation factor $\rho$ for $X_p$. We refer to Kivinen et al. [2006] for the formal definitions used for sparse / dense targets as well as for the experimental setting, which we have reproduced with the sole difference that the signal changes periodically each 1 000 iterations.

## Footnotes

[1]This result is stated as a bound on $D_{\varphi_q}(\boldsymbol{w}\|(\nabla\varphi_q)^{-1}(\nabla\varphi_q(\boldsymbol{w})+\boldsymbol{\delta}))$, which by the Bregman dual symmetry property is equivalent to a bound on $D_{\varphi_p}(\nabla\varphi_q(\boldsymbol{w})+\boldsymbol{\delta}\|\nabla\varphi_q(\boldsymbol{w}))$.

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

Table A2: Error($p$-LMS) - Error(DN-$p$-LMS) as a function of $t$ ($\in \{1, 2, \ldots, 50000\}$), $\boldsymbol{u}$ = dense, $(p, q) = (1.17, 6.9)$.

difference (%)

$\rho = 0.1$

$\rho = 0.2$

$\rho = 0.3$

$\rho = 0.4$

$\rho = 0.5$

$\rho = 0.6$

$\rho = 0.7$

$\rho = 0.8$

$\rho = 0.9$

$\rho = 1.0$

$\rho = 1.1$

$\rho = 1.2$

$\rho = 1.3$

$\rho = 1.4$

$\rho = 1.5$

$\rho = 1.6$

$\rho = 1.7$

Table A3: Error($p$-LMS) - Error(DN-$p$-LMS) as a function of $t$ ($\in \{1, 2, \ldots, 50000\}$), $\boldsymbol{u}$ = sparse, $(p, q) = (1.17, 6.9)$.

Table A4: Error($p$-LMS) - Error(DN-$p$-LMS) as a function of $t$ ($\in \{1, 2, ..., 50000\}$), $\boldsymbol{u}$ = dense, $(p, q) = (2.0, 2.0)$.

Table A5: Error($p$-LMS) - Error(DN-$p$-LMS) as a function of $t$ ($\in \{1, 2, \ldots, 50000\}$), $\boldsymbol{u}$ = sparse, $(p, q) = (2.0, 2.0)$.

difference (%)

$\rho = 0.1$  $\rho = 0.2$  $\rho = 0.3$  $\rho = 0.4$  $\rho = 0.5$  $\rho = 0.6$

$\rho = 0.7$  $\rho = 0.8$  $\rho = 0.9$  $\rho = \mathbf{1.0}$  $\rho = 1.1$  $\rho = 1.2$

$\rho = 1.3$  $\rho = 1.4$  $\rho = 1.5$  $\rho = 1.6$  $\rho = 1.7$

Table A6: Error($p$-LMS) - Error(DN-$p$-LMS) as a function of $t$ ($\in \{1, 2, \ldots, 50000\}$), $\boldsymbol{u}$ = dense, $(p, q) = (6.9, 1.17)$.

Table A7: Error($p$-LMS) - Error(DN-$p$-LMS) as a function of $t$ ($\in \{1, 2, ..., 50000\}$), $\boldsymbol{u}$ = sparse, $(p, q) = (6.9, 1.17)$.