[Reviews · NeurIPS 2016]

Reviewer 1

Summary

This paper aims to provide further understanding of how the *Bregman divergence* behaves under a certain type of transformation, in particular where the inputs are modified by a scaling function. The Bregman divergence is a generalization of a "distance measure" that arises in many areas of Machine Learning and is a key tool in understanding various aspects of convex analysis and optimization. The key result of the paper is to give full conditions under which the Bregman divergence is "well behaved" under scaling; that is under the mapping x --> x / g(x) where g() is some given function. The paper gives a central result along these lines, and then proceeds to describe three applications of the theorem: (a) a new way to compute multiclass density ratios, (b) an interesting approach to dual-norm mirror descent, and (c) some analysis tools for clustering on manifolds.

Qualitative Assessment

My overall assessment is mixed. Here are my high level points: 1. As someone who has worked a lot with Bregman divergences, I do find the main result to be quite appealing and intriguing. For general transformations, whenever you do map your inputs from one space to the other, it's typically never that simple to see what happens to the Bregman divergence on the resulting space. So this result seems to provide some clarity here in addition to being a generic tool. Also, the three applications provided are quite diverse and surprising, and these lend credibility to the claim that the result is broadly useful. 2. On the other hand, when you think about what the result is saying, it is fundamentally just a calculus identity. The authors are just showing that one has two different ways in which to compute "divergence from the linear approximation", under certain constraints on the function g(). When stated this way, it's more difficult to see this as an ML-ish result. 3. Finally, upon looking at the paper again, it does feel like the authors never really explain what the main theorem really means. The thing that gets me is that the transformation, x --> x / g(x), is just a weird one. The result seems to want to let g() be generic, but outside of normalization (i.e. dividing by a p-norm), how often does this type of transformation arise? And what do you do when g(x) = 0? The paper seems to want us to rely on the applications to answer these questions, but I would like further intuition as well. The authors may want to provide additional thoughts in their rebuttal.

Confidence in this Review

2-Confident (read it all; understood it all reasonably well)


Reviewer 2

Summary

This paper presents a scaled Bregman theorem which generalizes results in [Menon and Ong 2016]: conditions are given which characterize when "Bregman distortions" with possibly non-convex generators can be rewritten as a scaled Bregman divergence by transforming the inputs. This theorem is then applied to multiple density ratio estimation, a mirror descent algorithm for adaptive filtering, and clustering on curved manifolds.

Qualitative Assessment

The main result is interesting and useful, as shown by the applications, and the proofs are easy to follow. The writing in general was clear and concise, with surprisingly helpful figures. While the examples on pg 3 are helpful, I was hoping for more (or more verbose) discussion in the remark on line 81 of what the implications of (5) are for more general manifolds, as that is the focus of the paper. On lines 401,402, I believe B(.||.) should be D(.||.).

Confidence in this Review

2-Confident (read it all; understood it all reasonably well)


Reviewer 3

Summary

This paper gives a scaled Bregman theorem, which represents explicitly the Bregman divergence scaled by an affine function, as another Bregman "distortion" generated by the scaled convex function. Then the authors give three application scenarios to demonstrate the significance of the main theorem. First, the main theorem is applied to reduce the multiclass density-ratio estimation problem to class-probability estimation. Second, the main theorem is used to improve the p-LMS iteration, and the new iteration designed has an implicit regularization which p-LMS does not have. The explicit error estimate is given. Third, the main theorem is used to transform data for clustering problems on Riemannian manifolds with constant sectional curvature.

Qualitative Assessment

1. For the application scenario 1 (Section 3), I hope the authors state more on why the translation of density-ratio estimation to class-probability estimation, is a reduction. In particular, would this translation make the solver faster, or more stable? 2. For the plotting, it seems that the "hyperboloid" plotted has positive curvature, not negative. (top of page 6) 3. Some small typos, or language problems: Line 75, "and/or" is quite confusing. We suggest that the authors use more words to clarify the logic Line 146, "must the predict" should be "must then predict" or something else. Line 154, the "section" sign in (\SS6) is confusing.

Confidence in this Review

2-Confident (read it all; understood it all reasonably well)


Reviewer 4

Summary

This paper proves an equivalence theorem between certain Bregman distortions generated by possibly non-convex generators and scaled Bregman distances acting on inversely scaled data. Using this, the authors 1) prove a reduction of density ratio estimation to class probability estimation, 2) derive a projection-free dual norm mirror descent algorithm with regret gaurantee, and 3) demonstrate how to cluster data over Riemannian manifolds such as the sphere and hyperboloid using kmeans++. Experiments are presented validating the 3 above uses as well as the applicability of the main theorem. Other potentially fruitful uses of the theorem such as computing KL divergence with scaled natural parameters and nearest neighbor rules in computational geometry are discussed in the Appendix (E,F).

Qualitative Assessment

Based on the author's demonstrations using the scaled-Bregman theorem in 5 distinct domains, I expect many other people will be able to leverage the theorem in their work as well. The projection-free property of the DN-pLMS algorithm is particularly attractive and should see widespread use. Given the highly technical nature of the content, the writing is superb. The supplementary material includes extensive proofs and further experiments supporting the author's claims. Overall, this is a landmark paper. Strong accept. Comments: 1) Line 118: N not defined 2) Line 214: S introduced, but not clear how it is used in eqn (15) 3) Table 6 (L): y label says DM beats GKM, should be SKM beats Sk-means++? 4) Line 276: leverage? gain maybe 5) Line 282: 1%?

Confidence in this Review

2-Confident (read it all; understood it all reasonably well)


Reviewer 5

Summary

The authors show that Bregman distortions may be considered as Bregman divergencies applied to transformed data. They discuss three new applications of their theorem and provide experimental support for their theorem.

Qualitative Assessment

Well-structured paper, with intriguing results. I appreciate the potential for Theorem 1's wide applicability. With respect to the extension to the three novel areas: it seems that using the result pushes the heavier calculations into transforming the points properly, which may result in a problem that is just as difficult (if not more so) as considering the original problem. Nonetheless I believe it would be a helpful tool to keep under consideration when working in these areas.

Confidence in this Review

2-Confident (read it all; understood it all reasonably well)


Reviewer 6

Summary

The authors derive a scaled Bregman theorem that shows that a strictly generalized class of Bregman distortions can be re-expressed as Bregman divergences on transformed data. The authors also propose and empirically evaluate three applications of the theorem: a) reduction of multi-class density ratio estimation to multiclass probability estimation, b) updates for regularizerd stochastic mirror descent algorithm for adaptive filtering without involving projections, and c) clustering on curved manifolds.

Qualitative Assessment

Theorem 1 is a substantial generalization of existing results. The authors have further demonstrated the usefulness of their results in three concrete applications which are of independent interest. Minor: Line 44: \eta=r/g(r) is true only for the case of P(Y=1)=P(Y=-1)=1/2 Line 84: c.f. -> cf. Line 303: \phi,\phi' --> \phi_q,\phi'_q Eq 9: \Nabla \ell_t --> \Nabla \ell_t(w_t) Lemma 2: what are M and P_C? Appendix H: It would be instructional to see how the LHS of Lemma 2 result compare against Fig. 4. Suggestion: Appendix A after Appendix B might make for easier reading.

Confidence in this Review

2-Confident (read it all; understood it all reasonably well)